New holostean fishes (Actinopterygii: Neopterygii) from the Middle Triassic of the Monte San Giorgio (Canton Ticino, Switzerland)

López-Arbarello Adriana a.Lopez-Arbarello@lrz.uni-muenchen.de 1
Bürgin Toni 2
Furrer Heinz 3
Stockar Rudolf 4
1 SNSB—Bayerische Staatssammlung für Paläontologie und Geologie and GeoBio-Center, Ludwig Maximilian University , Munich , Germany
2 Naturmuseum , St. Gallen , Switzerland
3 Paläontologisches Institut und Museum, Universität Zürich , Zurich , Switzerland
4 Museo Cantonale di Storia Naturale , Lugano , Switzerland
Anquetin Jérémy
Electronic publication date: 2016 Jul 19
Publication date: 2016
Volume: 4
Electronic Location ID: e2234
Received 2016 May 6; Accepted 2016 Jun 17
Copyright: ©2016 López-Arbarello et al.
Copyright year: 2016
Copyright holder: López-Arbarello et al.
License: This is an open access article distributed under the terms of the Creative Commons Attribution License, which permits unrestricted use, distribution, reproduction and adaptation in any medium and for any purpose provided that it is properly attributed. For attribution, the original author(s), title, publication source (PeerJ) and either DOI or URL of the article must be cited.
License URL: https://creativecommons.org/licenses/by/4.0/

Keywords: Holostei, Halecomorphi, Intraspecific variation, Ginglymodi, Habitat partitioning, Durophagy, Trophic specialization, Middle Triassic

Funding: German Research Foundation DFG LO 1405/3-1 to 3-3 Museo Cantonale di Storia Naturale Max Kuhn The work of ALA was financed by the German Research Foundation (DFG LO 1405/3-1 to 3-3), the Canton Ticino through the Museo Cantonale di Storia Naturale (Lugano) and Max Kuhn (Uster). The funders had no role in study design, data collection and analysis, decision to publish, or preparation of the manuscript.

==============================
The new neopterygian genus Ticinolepis, including two new species T. longaeva and T. crassidens is described from Middle Triassic carbonate platform deposits of the Monte San Giorgio. The anatomy of this fish shows a mosaic of halecomorph and ginglymodian characters and, thus, the new taxon probably represents a basal holostean. During the latest Anisian to earliest Ladinian the two new species coexisted in the intraplatform basin represented by the uppermost Besano Formation, but only T. longaeva sp. nov. inhabited the more restricted basin represented by the Ladinian Meride Limestone (except for the Kalkschieferzone). The more widely distributed type species shows interesting patterns of intraspecific variation including ontogenetic changes and morphological variation over time. The second species presents anatomical features that strongly indicate a strictly durophagous diet. The different distribution of the species is interpreted as a result of habitat partitioning and different adaptability to palaeoenvironmental changes.

Introduction

The Neopterygii are the main group of actinopterygian fishes, and include the Teleostei, which account for almost half of modern vertebrate biodiversity. The origin of this important clade goes back to the Palaeozoic (Hurley et al., 2007; Sallan, 2014), but its most important radiation occurred in the early Mesozoic (Friedman, 2015). Ginglymodi, Halecomorphi and Teleostei are the three major clades currently recognized among crown neopterygians (Actinopterygii: Neopterygii). Whereas for much of the second half of the last century halecomorphs and teleosts were regarded as sister groups, to the exclusion of ginglymodians, recent morphological and molecular studies (e.g., Grande, 2010; Near et al., 2012) suggest that the latter form a monophyletic group together with halecomorphs: the Holostei. Ginglymodians and halecomorphs are extremely poorly represented today, including only seven living species of gars within the genera Lepisosteus and Atractosteus and a single living species of bowfin, Amia calva, respectively, contra more than 32,000 living species of teleosts (Eschmeyer & Fong, 2015). In the Triassic and Jurassic, however, ginglymodians and halecomorphs were the dominant groups of neopterygians and teleosts were just starting to diversify (Benton et al., 2013; Romano et al., 2016). Therefore, studies of early Mesozoic holosteans are necessary to understand the evolutionary history of this group and the processes through which teleosts outcompeted the holosteans in both marine and freshwater ecosystems during the late Mesozoic.

During the last decades, an increasing number of crown neopterygians have been described from Middle Triassic sediments of China (e.g., Tintori et al., 2007; Tintori et al., 2010; López-Arbarello et al., 2011; Wen et al., 2012; Xu & Wu, 2012; Chen et al., 2014; Xu, Zhao & Coates, 2014; Xu & Shen, 2015) and the Southern and Eastern Alps (e.g., Herzog, 2003; Bürgin, 2004; Herzog & Bürgin, 2005; Arratia & Herzog, 2007; Tintori & Lombardo, 2007; Tintori et al., 2007; Lombardo & Tintori, 2008; Lombardo, Tintori & Tona, 2012; López-Arbarello, Stockar & Bürgin, 2014), and several of these taxa are closely related to typical Jurassic neopterygians in Europe, indicating that the development of the Tethys played a very important role for the diversification of this group. The fossil assemblages from the Besano Formation and the Meride Limestone represent exceptional opportunities to study the evolution of life during the Middle Triassic, in particular for the time span between the latest Anisian and early late Ladinian. This time span is rarely as well represented and documented as in the Monte San Giorgio. In particular, biotas of latest Anisian—early Ladinian age are not known in the otherwise comparably rich Triassic successions of China (Benton et al., 2013; Tintori & Felber, 2015; Sun et al., 2016). Furthermore, the detailed geological information available for the Middle Triassic sequence of the Monte San Giorgio allows the placement of these fishes into a precise stratigraphic and palaeoenvironmental context (Rieber, 1973a; Bernasconi, 1994; Furrer, 1995; Röhl et al., 2001; Brack et al., 2005; Stockar, 2010; Stockar, Baumgartner & Condon, 2012; Stockar et al., 2013). Therefore, neither the Chinese nor the Alpine Triassic faunas overcome the value of the others (as awkwardly stated by Tintori & Felber, 2015), but they complement very efficiently. Studies of all these, as well as other Triassic faunas worldwide are necessary to explore the patterns of recovery of life during the Triassic after the Permo-Triassic mass extinction (Chen & Benton, 2012; Benton et al., 2013).

The present contribution deals with the detailed anatomical description, taxonomy and analyses of intra- and interspecific morphological variation of two new species of neopterygian fish from the upper Besano Formation (latest Anisian—earliest Ladinian) and the Meride Limestone (Ladinian) of the Monte San Giorgio (Canton Ticino, Switzerland). The specimens described here were previously identified as Archaeosemionotus sp., based on an erroneous concept of this genus, which was long considered to be a semionotid. The taxonomic status and phylogenetic relationships of the halecomorph Archaeosemionotus (Furidae) has recently been clarified (López-Arbarello, Stockar & Bürgin, 2014). This study is thus aimed to solve the taxonomy of the more than hundred specimens that have been excavated by the University of Zurich on the Swiss part of the Monte San Giorgio since 1924, and provisionally referred to that genus (Bürgin, 1998).

Geological setting

The Middle Triassic succession at the Monte San Giorgio (Figs. 1–2) starts with fluvio-deltaic deposits (Bellano Formation, Illyrian; Sommaruga, Hochuli & Mosar, 1997), unconformably overlying Lower Triassic transitional clastic deposits (Servino, Induan—Olenekian; Frauenfelder, 1916; Sciunnach, Gaetani & Roghi, 2015), in turn onlapping an erosional unconformity at the top of a Lower Permian volcanic basement. The upper Anisian sediments indicate the progressive transgression of a shallow epicontinental sea and the related expansion of carbonate platforms (San Salvatore Dolomite; Zorn, 1971) north of an emerged land area, which is nowadays covered by the Po Plain (Brusca et al., 1981; Picotti et al., 2007). During the latest Anisian and the Ladinian, although shallow-water sedimentation continued in the north, an intraplatform basin opened in the area of the Monte San Giorgio, which led to the deposition of the Besano Formation, the San Giorgio Dolomite, and the Meride Limestone (Rieber, 1973a; Bernasconi, 1994; Furrer, 1995; Röhl et al., 2001).

Figure 1 Location of the fossil sites.

Geological map of the Monte San Giorgio area showing the Middle Triassic carbonate sequence and the location of the excavation sites in which Ticinolepis gen. nov. has been collected (after Stockar & Kustatscher 2010, modified).

Figure 2 Stratigraphic chart of the Middle Triassic sequence of the Monte San Giorgio.

Middle Triassic stratigraphic units of the Monte San Giorgio area and occurrences of Ticinolepis gen. nov. Stratigraphy after Commissione scientifica transnazionale Monte San Giorgio 2014, modified. U-Pb ages after Mundil et al. (2010) and Stockar, Baumgartner & Condon (2012).

The famous vertebrate fossils (reptiles and fishes) of the Monte San Giorgio were discovered in the middle part of the Besano Formation (“scisti bituminosi”) near Besano on the Italian side of Monte San Giorgio in the middle of the 19th century and were described first by Cornalia (1854), Stoppani (1857), Curioni (1863) and Bassani (1886). De Alessandri (1910) published the first monograph on the fish fossils. Most fossils were collected in material coming out from galleries of mines near Besano and Serpiano, where bituminuous shales have been exploited. The University of Zurich started first systematic excavations near Serpiano on Swiss territory (CaveTre Fontane and Val Porina mines) in 1924 (Peyer, 1944), and a standard section of the Besano Formation was studied in detail from 1950 to 1968 in the excavation P. 902/Mirigioli (“Grenzbitumenzone” in Rieber, 1973b; Kuhn-Schnyder, 1974). The rich and well preserved reptile fossils (ichthyosaurs, sauropterygians, protorosaurs and rauisuchians) were described almost completely (references in Furrer, 2003); however only a small part of the large and high diverse fish material has been published (Schwarz, 1970; Rieppel, 1980; Rieppel, 1981; Rieppel, 1982; Rieppel, 1985; Rieppel, 1992; Bürgin, 1990, Bürgin, 1992; Bürgin, 1999a; Bürgin, 2004; Romano & Brinkmann, 2009; Maxwell et al., 2015).

The Besano Formation (“Grenzbitumenzone” of Frauenfelder, 1916) consists of an alternation of dolomites and black shales of up to 16 m in thickness, including the Anisian-Ladinian boundary in its uppermost part (Fig. 2; Brack et al., 2005). A volcanic-ash layer six meters below this boundary yielded a U-Pb minimum age of 242.1 ± 0.6 Myr (Mundil et al., 2010). The lower Besano Formation with laminated to stromatolitic dolomites, very thin organic rich black shales and disarticulated vertebrate fossils was deposited in a shallow restricted carbonate platform environment (Röhl et al., 2001). The middle part with prominent black shales in between dolomitic mud- to packstones and a high diverse and well preserved vertebrate fauna documents a slightly deeper intraplatform basin with euxinic and anoxic conditions. The upper Besano Formation with dominant laminated dolomites, thin black shales and well preserved but less diverse vertebrates, was deposited again in a shallow restricted carbonate platform environment (Röhl et al., 2001). The fishes included in this study have been collected from 13 different dolomite and black shale levels (beds) within the uppermost part of the Besano Formation (earliest Ladinian, E. curionii Ammonoid Zone), some fragmentary specimens referable to the new genus were also found in the lower part of the upper Besano Formation (latest Anisian, N. secedensis Ammonoid Zone; Fig. 3).

Figure 3 Stratigraphy of the Besano Formation at the Monte San Giorgio locality P. 902/Mirigioli with position and number of the described specimens of Ticinolepis gen. nov.

The material from the excavation at the Val Porina mine is indicated on the right side.

The Besano Formation grades upwards into the San Giorgio Dolomite. The overlying 400–600 m thick Meride Limestone (Wirz, 1945) begins with the lower Meride Limestone, which is up to 150 m thick and is interpreted as a sequence of lime mud turbidites (Furrer, 1995). The Lower Meride Limestone includes three fossil vertebrate levels: Cava inferiore, Cava superiore and Cassina beds, each yielding different vertebrate assemblages (e.g., Sander, 1989; Furrer, 1995; Bürgin, 1999b; Stockar & Renesto, 2011) and consisting of finely laminated limestones and marls with intercalated volcanic ash layers.

The 1.50 m thick fossiliferous sequence of black mudstone, laminated limestone and dolomite of the Cava inferiore beds was discovered and studied first in 1927 by the palaeontologists from the University of Zurich at the locality Acqua del Ghiffo, northwest of Meride (Peyer, 1932). Many small and well-preserved pachypleurosaurs, a few larger nothosaurs, and some actinopterygian fishes were found. Additional material was excavated in 1995/96, including Saurichthys, Ctenognathichthys, Peltopleurus, Peripeltopleurus, Habroichthys, Eosemionotus, Placopleurus, Legnonotus, Eosemionotus and some specimens included in this study (Archaeosemionotus sp. in Bürgin, 1999a; Bürgin, 1999b; Furrer, 1999a). Many pachypleurosaurs, a few juvenile nothosaurs, and some actinopterygian fishes (Saurichthys, Besania, Eosemionotus, and one fish included in this study) were also found in the 10 m thick Cava superiore beds with their finely laminated limestone sequence, excavated in 1928 and 1997–2004 by teams from the University in Zurich at the locality Acqua del Ghiffo (Peyer, 1931; Bürgin, 1999b; Furrer, 1999b; Furrer, 2003).

The Cassina beds in the Canton Ticino, Switzerland, were discovered in 1933 by a team of the University of Zurich, which carried out several subsequent excavations in 1933, 1937, between 1971 and 73, and again in 1975 (for a review see Stockar 2010). Many well preserved marine reptiles, actinopterygian fishes and terrestrial plants were found and documented bed by bed. Whereas the reptiles (pachypleurosaurs, nothosaurs, protorosaurs) were described in detail (see Furrer 1995; Hänni 2004), the actinopterygians (Saurichthyidae, Peltopleuridae, Macrosemiidae, and the fishes described herein) were only partly prepared and described (Rieppel, 1985; Bürgin, 1999a). In 2006, the Museo Cantonale di Storia Naturale (Lugano) opened a new excavation at the type locality (Fig. 1), to investigate a surface of about 40 square meters bed by bed through the whole fossiliferous succession. In the new site, the Cassina beds reach a thickness of almost 3 m and grade upwards into thick-bedded dolomitic limestones and dolomites. The studied section (Fig. 4; Stockar, 2010) mainly consists of interbedded finely laminated and organic-rich limestones with intercalated thicker micritic limestones and volcanic ash layers. A 5-cm-thick volcanic ash layer resulted in a U-Pb age of 240.63 ± 0.13 Myr (Stockar, Baumgartner & Condon, 2012). Along with sauropterygian remains (Stockar & Renesto, 2011), land plant remains (Stockar & Kustatscher, 2010) and quasi-anaerobic foraminiferal faunas (Stockar, 2010) the new excavations brought to light many complete and well-preserved specimens of several actinopterygian fishes, including the taxa Saurichthys, Besania, Eosemionotus, Peltopleurus, Eoeugnathus (Renesto & Stockar, 2009; Stockar, 2010; R Stockar, 2014, unpublished data), and the new basal neopterygian genus described herein. Coupled with widespread oxygen-depletion excluding benthic scavengers, rapid coating of skeletons by benthic bacterial mats and high sedimentation rate played the key role in protecting the carcasses from decay and in holding skeletal elements together (Stockar et al., 2013; Beardmore & Furrer, 2016a; Beardmore & Furrer, 2016b). The top of the lower Meride Limestone is defined by a very discontinuous dolostone horizon (“Dolomitband”; Frauenfelder, 1916) resulting from late diagenetic dolomitization cutting across the stratification of the Meride Limestone (Stockar et al., 2013).

Figure 4 Distribution of Ticinolepis longaeva gen. et sp. nov. in the Cassina beds.

Detailed sedimentological log of the upper part of the Cassina beds excavated by the MCSN (after Stockar & Kustatscher, 2010, updated).

The overlying upper Meride Limestone is a sequence of alternating well-bedded micritic limestone and marlstone. Its lowermost part includes a newly discovered fossil vertebrate level, the Sceltrich beds (Stockar et al., 2013; Stockar & Garassino, 2013). After a first exploration in 2010 yielding the first fossils from this horizon, two small bed by bed excavations on a surface of around 6 and 10 square meters, respectively, were started in 2012 by the Museo Cantonale di Storia Naturale (Lugano). The site is located on the northern bank of a small creek (Valle di Sceltrich; Fig. 1), northwest of the village of Meride. The fossiliferous interval consists of a 30 cm thick sequence of prevailing organic-rich laminated limestone intercalated between thick-bedded marly limestone. So far, the excavation carried out in the Sceltrich beds yielded a rich vertebrate fossil fauna (mostly articulated fishes and rare sauropterygian reptile bones and teeth), along with invertebrate fossils (bivalves, gastropods, crustaceans) and terrestrial plant remains. Benthic microbial activity under lower dysoxic to anoxic bottom-water conditions accounts for the microfabrics observed in the laminated limestone and for the exquisite preservation of the vertebrate fossils (Stockar et al., 2013).

Further up in the stratigraphic section, only one well-preserved, but fragmentary specimen studied here (MCSN 5827) was found in 1995 by David Cook (University of Zurich) on the right (southern) side of Val Serrata (path from Meride to Riva San Vitale) on a limestone slab in the scree coming from the Upper Meride Limestone, but below the Kalkschieferzone. This is the youngest specimen of the new species identified so far.

Material and Methods

The fossil material was mechanically prepared with the aid of vibrotools and sharpened steel needles. The specimens were studied under a stereomicroscope Leica Wild MZ6 and M80 equipped with a camera lucida. The drawings were made directly on the fossil and most of the photographs were taken with a Nikon D40 digital camera equipped with a Nikon AF-S micro 60 mm objective. Detailed photographs of some anatomical structures were taken with the same Nikon camera attached to the microscope via a phototubus.

Skull bones cephalic sensory canals are named according to the use of most authors in actinopterygians. The bones carrying the infraorbital sensory canal anterior to the orbit are referred to as ‘anterior infraorbitals’ following Wenz (1999), Wenz (2003) and López-Arbarello & Codorniú (2007). Fringing fulcra are named according to Patterson (1982). Scutes, unpaired and paired basal fulcra are identified according to López-Arbarello & Codorniú (2007). The relative position of the fins and the scale counts are expressed in a pterygial formula where D, P, A, and C indicate the number of scale rows between the first complete row behind the pectoral girdle and the insertion of the dorsal, pelvic, anal, and caudal fins respectively, and T is the total number of scale rows between the pectoral girdle and the caudal inversion (Westoll, 1944).

The systematic nomenclature follows López-Arbarello (2012) for Ginglymodi, López-Arbarello, Stockar & Bürgin (2014) for Halecomorphi, and Sferco, López-Arbarello & Báez (2015) for Teleostei. Accordingly, the informal term ‘gars’ correspond to the Lepisosteoidei of López-Arbarello (2012), which is equivalent to the Lepisosteiformes of Grande (2010), and the term ‘teleosts’ correspond to the total clade Teleostei as defined by De Pinna (1996).

Figure 5 Holotype specimens of the two species of Ticinolepis gen. nov.

Ticinolepis gen. nov. Photographs of the holotype specimens of the two species included in the genus. (A) T. longaeva sp. nov., type species of the genus, MCSN 8072 (holotype; SL 12 cm) preserved in right lateral view. (B) T. crassidens sp. nov., PIMUZ T 273 (SL 9 cm), preserved in left lateral view. Scale bars = 2 cm.

The signs attached to the entries in the synonymy list follow Matthews (1973).

The electronic version of this article in Portable Document Format (PDF) will represent a published work according to the International Commission on Zoological Nomenclature (ICZN), and hence the new names contained in the electronic version are effectively published under that Code from the electronic edition alone. This published work and the nomenclatural acts it contains have been registered in ZooBank, the online registration system for the ICZN. The ZooBank LSIDs (Life Science Identifiers) can be resolved and the associated information viewed through any standard web browser by appending the LSID to the prefix http://zoobank.org/. The LSID for this publication is: urn:lsid:zoobank.org:pub:64547327-C8AE-4CD3-8776-F3CA89E1744E. The online version of this work is archived and available from the following digital repositories: PeerJ, PubMed Central and CLOCKSS.

Results

Systematic palaeontology

ACTINOPTERYGII Cope, 1887	
NEOPTERYGII Regan, 1923	
HOLOSTEI Müller, 1844 sensu Huxley, 1861	
Holostei incertae sedis	
Genus Ticinolepis gen. nov.	
urn:lsid:zoobank.org:act:0DDFF9EC-8861-42A5-87A7-744153193A42	
(Fig. 5)	

v. 1998 Archaeosemionotus sp. Bürgin: Fig. 8H, Table p. 7

v. 1999a Archaeosemionotus sp. Bürgin: Fig. 6, Appendices 1–2

v. 1999b Archaeosemionotus sp. Bürgin: Fig. 2

v. 2010 Archaeosemionotus sp. Stockar: Figs. 6A–6C

Etymology. The name Ticinolepis recalls the Ticino, which is the southernmost canton of Switzerland and encompasses the type locality of the type species of the genus.

Type species. Ticinolepis longaeva sp. nov.

Additional species. Ticinolepis crassidens sp. nov.

Diagnosis. Ticinolepis gen. nov. differs from all other neopterygian taxa in the following combination of characters, including unique features (indicated with ‘*’): large nasals; largest infraorbital bone at the centre of the ventral rim of orbit*; mosaic of suborbital bones; large subtriangular posttemporal bones tightly attached to the extrascapulars and meeting at dorsal midline; presupracleithrum present; thick ganoin patches on fringing fulcra*; ventrum covered with very shallow scales.

Remarks. The specimens here referred to the new genus Ticinolepis have previously been erroneously identified in the genus Archaeosemionotus, the taxonomy and phylogenetic relationships of which have been revised by López-Arbarello, Stockar & Bürgin (2014). Very different from Ticinolepis, Archaeosemionotus presents three very large infraorbital bones forming the ventral margin of the orbit, with the first and third of them being the largest bones in the circumborbital series; there are no anterior infraorbitals; the cheek is covered with only two large suborbital bones, with a little third suborbital between them at their posterior margins; the maxilla is much longer, reaching the level of the posterior border of the orbit; and there is a large gular plate. According to the cladistic analysis performed by López-Arbarello, Stockar & Bürgin (2014), Archaeosemionotus is an halecomorph closely related to the early Middle Triassic (Anisian) Robustichthys luopingensis Xu, Zhao & Coates, 2014, from China and the Late Jurassic (Kimmeridgian) Ophiopsis muensteri Agassiz, 1833, from Germany.

The specimens MCSN 3012, 3013 and PIMUZ T 4963 from the Kalkschieferzone of the Meride Limestone, which have been referred to Arcahaeosemionotus sp. in Bürgin (1995), do not represent any species of Ticinolepis gen. nov.

Among other features, the presence of a movable maxilla with a well-developed anterior articular process, a composite coronoid process of the mandible, and an interopercle clearly indicates that Ticinolepis is a member of the crown group Neopterygii (Patterson, 1973; Friedman, 2015). The presence of a small rostral bone without internasal lamella, a tube like canal bearing anterior arm on the antorbital, a supraangular bone, and a well-developed nasal process in the premaxillae allow the classification of the new genus in the Holostei (Grande, 2010).

Among the diagnostic characters of the genus, features indicating sister group relationship for the two new species are the largest infraorbital bone at the centre of the ventral rim of orbit and the presence of thick ganoin patches on the fringing fulcra. In most other holosteans the largest bone in the circumborbital series is: (1) one of the supraorbitals in Lepidotes or Scheenstia (López-Arbarello, 2012); (2) a supraorbital or the dermosphenotic in lepisosteid gars (Grande, 2010), Isanichthys (Cavin & Suteethorn, 2006; Deesri et al., 2014) or Thaichthys (Cavin, Deesri & Suteethorn, 2013); (3) the infraorbital placed at the posteroventral corner of the orbit (jugal of some authors) in several taxa like Araripelepidotes (Maisey, 1991), Paralepidotus (Tintori, 1996), Sangiorgioichthys (Tintori & Lombardo, 2007; López-Arbarello et al., 2011; Lombardo, Tintori & Tona, 2012; Chen et al., 2014), Semiolepis (Lombardo & Tintori, 2008), Semionotus bergeri (López-Arbarello, 2008) among ginglymodians, or in amiiforms (Grande & Bemis, 1998) or Robustichthys (Xu, Zhao & Coates, 2014), among halecomorphs; (4) either the infraorbital placed at the posteroventral or the anteroventral corner of the orbit (or lacrimal) in Amiopsis or ionoscopiform halecomorphs (Grande & Bemis, 1998; López-Arbarello, Stockar & Bürgin, 2014); or (5) one of the anterior infraorbitals in Semionotus elegans (Olsen & McCune, 1991) or the macrosemiids (Bartram, 1977). In Ticinolepis thus differing from all other crown neopterygian (including basal teleosts), the largest bone in the circumborbital series is the infraorbital forming the ventral margin of the orbit, which is between two smaller infraorbitals placed at the antero- and posteroventral corners of the orbit (Figs. 6A–6B). Also unique of the new genus is the presence of thick ganoin ovoid patches on the distal end of the fringing fulcra (Figs. 6C–6D), which are so far unknown in other crown neopterygians.

Figure 6 Diagnostic features of Ticinolepis gen. nov.

Features uniquely derived in Ticinolepis gen. nov. Largest infraorbital bone at the centre of the ventral rim of orbit: (A) T. longaeva sp. nov., line drawing of MCSN 8072 (holotype). (B) T. crassidens sp. nov., line drawing of PIMUZ T 277. Thick ganoin patches (black arrows) on the fringing fulcra at the ventral margin of the caudal fin in: (C) T. longaeva sp. nov., photograph of MCSN 8009; (D) T. crassidens sp. nov., photograph of PIMUZ T 256. Scale bars = 5 mm.

Figure 7 Skull of the holotype of Ticinolepis longaeva gen. et sp. nov.

Ticinolepis longaeva gen. et sp. nov., skull of MCSN 8072 preserved in right lateral view. (A) Photograph. (B) Line drawing. The ascending and basipterygoid processes of the parasphenoid are indicated with green and red circles, respectively. Arrows point to the rounded ventral process of the posttemporal. Scale bars = 1 cm.

Figure 8 Ticinolepis longaeva gen. et sp. nov., MCSN 8007.

(A) Overview of the disarticulated skull; scale bar = 1 cm. (B) Detailed photograph of the vomers; scale bar = 2 mm. (C) Detailed photograph of the left dermopterotic preserved in medial view; scale bar = 2 mm.

Figure 9 Ticinolepis longaeva gen. et sp. nov., skull of MCSN 5827 preserved flattened in dorsal view.

(A) Photograph. (B) Line drawing. Scale bars = 1 cm.

Figure 10 Ticinolepis longaeva gen. et sp. nov., skull of MCSN 8469 preserved in dorso-lateral (left) view.

(A) Photograph. (B) Line drawing. Arrow points to the small rounded bone between the left frontal and nasal bones. Scale bars = 1 cm.

Figure 11 Ticinolepis longaeva gen. et sp. nov. PIMUZ T 486.

(A) Overview of the disarticulated skull; scale bar = 2 cm. (B) Line drawing of the hyomandibula; scale bar = 5 mm. (C) Line drawing of the articulated posttemporal and supracleithrum; scale bar = 5 mm.

Figure 12 Ticinolepis longaeva gen. et sp. nov., MCSN 8009 preserved in left lateral view.

(A) Overview of the complete specimen; scale bar = 2 cm. (B) Skull; scale bar = 1 cm. (C) Caudal fin; scale bar = 1 cm. Arrows point to the rounded ventral processes of the posttemporal bones.

Figure 13 Further anatomical details of Ticinolepis longaeva gen. et sp. nov.

Ticinolepis longaeva gen. et sp. nov. (A–C) Symplectic-quadrate-quadratojugal complex: (A) Right complex fully articulated, quadratojugal fused to quadrate, all elements ventrolaterally compressed and exposed in lateroventral view in PIMUZ T 4987; (B) Symplectic disarticulated, clockwise rotated and displaced, all elements exposed in medial view in MCSN 8009; (C) Interpretative reconstruction of the three bones articulating in median view; the arrow points to the posterodorsally directed rounded shelf, on which the symplectic was probably able to slide. (D) Right premaxilla preserved in anterodorsal view in MCSN 8007; (E) Left premaxilla preserved in median view in MCSN 8009. Scale bars = 2 mm.

Figure 14 Ticinolepis longaeva gen. et sp. nov., skull of MCSN 8001 preserved in left lateral view.

(A) Overview of the partially articulated skull preserved; scale bar = 1 cm. (B) Line drawing of the upper and lower jaws; scale bar = 5 mm. Arrow points to the rounded ventral process of the posttemporal.

Figure 15 Ticinolepis longaeva gen. et sp. nov., median fins.

(A) Dorsal fin in MCSN 8073. (B) Anal fin in MCSN 8073. (C) Caudal fin in MCSN 8475. Scale bars = 5 mm.

Ticinolepis longaeva sp. nov.	
urn:lsid:zoobank.org:act:31304A6C-ADF6-4C37-92F2-62E505B2823C	
(Figs. 5A, 6A, 6C, 7–15; Appendix S1: Figs. 1–7)	

p v. 1998 Archaeosemionotus sp. Bürgin: Fig. 8H, Table p. 7

p v. 1999a Archaeosemionotus sp. Bürgin: Appendix 1

v. 1999b Archaeosemionotus sp. Bürgin: Fig. 2

v. 2010 Archaeosemionotus sp. Stockar, Figs. 6A–6C

Holotype. MCSN 8072 (Fig. 5A).

Paratypes. MCSN 8001, 8007–9, 8011, 8073, 8077, 8083–4, 8086, 8412–13, 8415; PIMUZ T 1789, 4785, 4788, 4789, 4790, 4987.

Type locality. Cassina, Monte San Giorgio (Meride, Cantone Ticino, Switzerland; Fig. 1).

Type horizon. Cassina beds of the Meride Limestone, early Ladinian (uppermost P. gredleri Ammonoid Zone or transition interval between P. gredleri and P. archelaus Ammonoid Zone), Middle Triassic (Figs. 2 and 4).

Etymology. The species name longaeva (= long-lived) derives from the latin ‘longaevus’, which is a compound of ‘longus’ (= long, prolonged) and ‘aevus’ (later ‘aevum’ = age, temporal duration). It refers to the long stratigraphic range of this species compared with the other species of the genus.

Referred material. From the Meride Limestone: MCSN 4974 from the Cava inferiore beds (Acqua del Ghiffo); PIMUZ T 3015 from the Cava superiore beds (Acqua del Ghiffo); MCSN 8317, 8351, 8417, 8424, 8427, 8441, 8446, 8448, 8467, 8468, 8469, 8470, 8471, 8475, and 8476 from the Sceltrich beds (Sceltrich valley); MCSN 5827 from the upper Meride Limestone, found on the path from Meride to Riva San Vitale, right flank of Val Serrata.

From the Besano Formation: PIMUZ T 226, 252, 275, 276, 362, and 486 from the uppermost Besano Formation (P. 902/Mirigioli/); PIMUZ T 2848, 2944, 2999, 3269, and 4783 from the uppemost Besano Formation (Val Porina mine). The Bender collection of the Natural History Museum in London includes two specimens (NHMUK P 19351 and P 19355), which can also be referred to this new species (T Bürgin, pers. obs., 1989).

Distribution. The species is known from two different formations within the Middle Triassic sequence of the Monte San Giorgio in Switzerland: eight different levels within the uppermost part of the Besano Formation (early Ladinian, E. curionii Ammonoid Zone; Fig. 2), and five different stratigraphic levels within the Meride Limestone (early–late Ladinian, P. gredleri and P. archelaus Ammonoid Zones): the Cava inferiore beds, the Cava superiore beds, the Cassina beds (type locality), the slightly younger Sceltrich beds and a higher level in the upper part of the formation (?late Ladinian); but not from the Kalkschieferzone (Fig. 2).

Diagnosis. Species of Ticinolepis differing from the other species of this genus in the following combination of characters (potential apomorphies are indicated with ‘*’): nasals extremely broad, the posterior border as broad as anterior border of frontal and anterior supraorbital together*; 5 infraorbital bones; largest infraorbital fan-shaped*; coronoid and marginal teeth of the jaws pencil-like, graceful; 11 principal caudal fin rays below the lateral line; numerous longitudinal rows of scales, 14–22 above the lateral line (varying according to the size of the specimens); marginal row of body lobe with seven scales; second inverted row is the longest, longer than third row, made up of 11–14 scales.

Pterygial formula:

D20–21P6–7A16–18C30–32T37–38.

Anatomical description

Ticinolepis longaeva is a medium size, up to c. 25 cm SL fish with fusiform body (Fig. 5A; most specimens between 10 and 20 cm SL). Most of the fossils have partially disarticulated skulls and/or have some degree of torsion and deformation and, thus, it is not possible to take precise measurements. However, some morphometric proportions are estimated. The skull is relatively large, approximately as deep as the body and the skull length represents 30–40% of the SL. The orbit is rather small, only about 20% of the length of the head. The preorbital length to orbital length 134–159%; preorbital length to postorbital length 61–83%; the maximal body depth is about 35% of the SL. The triangular dorsal fin places in the middle of the body between the pelvic and anal fins, originating at about 60% of the SL. The pelvic fins originate at approximately the same level of the origin of the dorsal fin, and the anal fin originates at c. 75% of the SL.

The vertebral column of T. longaeva is covered with the thick ganoid scales in all the specimens and, thus, its anatomy remains unknown. However, at least six well ossified ribs are exposed in MCSN 8475 (Appendix S1: Fig. 1) indicating that the vertebral centra were at least partially ossified.

Braincase, parasphenoid and vomers

The braincase is partially exposed in several specimens, but only a few bones are recognizable. MCSN (8317) shows an approximately circular prootic with only one large foramen for the hyomandibular trunk and the jugular vein (Allis, 1897; Bjerring, 1972; Bjerring, 1977); posterior to this foramen there is a deep excavation corresponding to the subtemporal fossa (Appendix S1: Fig. 2). The sphenotic is pyramidal and sutures to the dermopterotic posteriorly. The orbitosphenoid is preserved in MCSN 8072 (Fig. 7). The bone has a rounded convex dorsal border and a notch in the anterior border. Comparable notches are visible in the orbitosphenoid of Isanichthys lertboosi (Deesri et al., 2014: Figs. 7–8), gars (Grande, 2010) and amiiforms (Grande & Bemis, 1998).

The parasphenoid has well-developed basipterygoid and ascending processes (= processus ascendens anterior and posterior, respectively—see ‘Discussion’; MCSN 8008, 8072, 8086, 8317, 8351; Fig. 7, Appendix S1: Figs. 3–4). The orbital portion of the parasphenoid is broader than the postorbital portion and there are no teeth and no foramina. On the dorsal surface there is a median flattened strong ridge extending forward from the ascending process widening and becoming shallower in anterior direction to completely form the dorsal surface of the parasphenoid (MCSN 8008, 8086; Appendix S1: Fig. 4).

The paired vomers are disarticulated and nicely exposed in MCSN 8007 and PIMUZ T 4785 (Fig. 8, Appendix S1: Fig. 5A). Each vomer has a relatively large rectangular anterior head and a posterior process that sutures with the parasphenoid. The length of the anterior head represents almost half of the total length of the bone and its width is about 1.3 times the maximal width of the posterior process. The vomerine dentition is heterogeneous. There is a marginal row of four small teeth followed by a median patch of four irregularly arranged slightly larger teeth and a posterior row of two large fangs.

Skull roof and snout

The dermal bones of the skull roof are ornamented with numerous tubercles and short ridges covered with ganoin. The frontals cover most of the skull roof and both frontals are of similar size except in MCSN 8072, in which they are asymmetrical, the left frontal being somewhat larger (Figs. 7–10; Appendix S1: Figs. 3–5). The two frontals do not extend far beyond the anterior margin of the orbit, but they do posteriorly, covering the anterior half of the temporal region of the skull, so that the postorbital portion of the frontal is about 35–40% of the total length of the bone. The bones are rather wide; their length is between 2.5 and 3 times its maximal width, which is in the temporal portion approximately at the level of the anterior border of the dermopterotic. The interorbital constriction is nicely curved and deep and the frontals are expanded again anteriorly. The anterior border of the frontals is oblique in posterolateral to anteromedial direction. The posterior border of the frontal is rounded to almost rectangular and the suture between the frontals is mainly smooth though it forms one or two strong indentations approximately at the level of the posterior rim of the orbit. The form of this indentation varies from clearly indenting processes (MCSN 8009, 8072, PIMUZ T 4785; Figs. 7 and 12, Appendix S1: Fig. 5A) to an almost quadrangular step (MCSN 8007, PIMUZ T 4788; Fig. 8, Appendix S1: Fig. 5B). The path of the supraorbital sensory canal is preserved in the lateral half of each frontal and is marked by a strong ridge in the medial surface of the bone (MCSN 8007, 8008). It is straight and close to the central longitudinal axis of the frontal in the anterior half of the bone and it curves laterally approaching the lateral border and the dermosphenotic at the beginning of the temporal portion of the bone. Only the posterior half to two thirds of the frontals is ornamented with tubercles and short ridges resembling dropping water radiating from the postorbital curvature of the path of the supraorbital sensory canal.

Both quadrangular parietals are of comparable size and their length is between 25 and 30% of the length of the frontals. The suture between both parietals is straight and smooth in some specimens (MCSN 8072; Fig. 7), but forms a posterior indentation in others (MCSN 8009, PIMUZ T 4788; Fig. 12, Appendix S1: Fig. 5B). The parietals are completely and densely ornamented and the tubercles and ridges are approximately radiating from the posterolateral region of the bone. Laterally, the parietals and frontals suture with the dermopterotics, which are elongate, more than two times longer than deep, narrowing anterolaterally following the posterior edge of the frontal. The dermopterotics are about 1. 5 times longer than the parietals extending forwards lateral to the frontals for 40–45% of their total length in the larger specimens (MCSN 8009, 8072, PIMUZ T 4987; Figs. 7 and 12, Appendix S1: Fig. 6). In this anterior portion, the medial margin of each dermopterotic forms a step, which is overlapped by the frontal (MCSN 8072; Fig. 7). In medial view, the dermopterotics are longitudinally traversed by a well-developed descending lamina, which runs mostly through the middle of the bone and bends laterally rather abruptly close to the posterior border (MCSN 8007, 8008; Fig. 8, Appendix S1: Fig. 4). On the dorsolateral surface, only the posteromedial portion of the dermopterotics is ornamented and the posterior portion of each dermopterotic is traversed by a transversal groove housing the middle pit line. There is no groove for the middle pit line in the parietals. The supraorbital sensory canal does not enter the parietals.

Parietals and dermopterotics articulate posteriorly with a single pair of large extrascapular bones. Each extrascapular is sub-triangular, but only slightly narrowing towards the midline where it meets its counterpart forming a smooth suture. The extrascapulars are tightly bound to the large and similarly shaped posttemporals, and both bones are also similarly and densely ornamented.

There is a large and broad pair of nasal bones and a median rostral. The three bones articulate with each other covering the ethmoidal region completely (MCSN 8072, 8073, 5827; Figs. 7 and 9, Appendix S1: Fig. 7). The nasals are very broad posteriorly, with the posterior border as broad as the anterior borders of the frontals and anterior supraorbital together in large specimens (MCSN 5827, 8469; Figs. 9–10). From this broad posterior portion, the nasals narrow anteriorly and bent laterally at the anterior end. Thus, in the anterior region, the median border of the nasals is convex and the lateral border is deeply concave, probably surrounding the anterior external nares. The shape of the large nasals does not provide any clear indication of the position of the posterior nares, but they were probably placed between the nasals and the frontals as in Amia (Grande & Bemis, 1998). An excavation on the posteromedial corner of the nasals is present in medium size individuals (e.g., MCSN 8007, 8009, 8072, 8413; Figs. 7–8 and 12), but it is not present in larger specimens. In some of the largest specimens one pair of small rounded bones are placed, one on each side, between the frontal and the nasals on the area where the notch is placed in the smaller fishes (MCSN 5827, 8469; Figs. 9–10). In one of the largest specimens, this small bone is ornamented with a ganoin patch (MCSN 8469; Fig. 10). The dorsal surface of the nasals is traversed by a longitudinal ridge, which enclosed the anterior portion of the supraorbital sensory canal and bents laterally towards the antorbital in the anterior portion of the nasal. There are a few scattered tubercles ornamenting the nasals. The rostral is roughly tubular, fitting the curve formed by the two articulated nasals, and it is concave anteriorly (MCSN 5827, 8001, 8007, 8073, 8413, PIMUZ T 4987; Figs. 8–9 and 14; Appendix S1: Figs. 6–7). There are no lamellar expansions underlying the nasals and there is a tube for the ethmoidal commissure.

Circumborbital series and cheekbones

The circumborbital series consists of supraorbitals, dermosphentoic, infraorbitals, anterior infraorbitals and antorbital bones. There are two (MCSN 8073, 8469, PIMUZ T 4783, 4788; Fig. 10, Appendix S1: Figs. 5B and 7) or three (MCSN 5827, 8009, 8072; Figs. 7, 9 and 12) supraorbital bones. The supraorbital bones are ornamented with tubercles and short ridges. The dermosphenotic is a little larger than the last supraorbital and strongly sculpture with ridges and shelves, probably following the paths of the sensory canals (MCSN 8072; Fig. 7). It is tightly attached to the sphenotic at the posterodorsal corner of the orbit in several specimens, but it is not incorporated into the skull roof. The junction between the supraorbital, infraorbital and temporal canals most probably occured in the dermosphenotic. In medial view the dermosphenotic is traversed by a strong ridge (MCSN 8007, 8008; Fig. 8C, Appendix S1: Fig. 4).

There are three infraorbital bones forming the ventral rim of the orbit and one (e.g., MCSN 5827, PIMUZ T 4987) or two (e.g., MCSN 8009, 8073) small, almost tubular infraorbitals in the posterior border of the orbit. The first infraorbital bone is subrectangular, somewhat longer than it is deep, and does not contact the first supraorbital so that the orbit is not closed anteriorly (MCSN 8072, 5827; Figs. 7 and 9). The second infraorbital is notably the largest in the series and it is longer than it is deep, expanded ventrally in an almost fan-shaped lamina, which is distinctly traversed by ventrally directed grooves radiating from a main longitudinal groove related to the infraorbital sensory canal. Similar main longitudinal and minor radiating grooves are more poorly preserved in the first infraorbital bone. The second infraorbital alone makes most of the ventral rim of the orbit and it is followed by a small infraorbital, which is forming the posteroventral corner of the orbit. This third infraorbital is only slightly larger than the infraorbitals forming the posterior border of the orbit, which are the smallest in the series, subrectangular, slightly deeper than long. The infraorbitals are ornamented with a few scattered tubercles.

Anterior to the first infraorbital there are two anterior infraorbitals. The large nasals are displaced over the anterior infraorbitals in most of the specimens. However, the anterior infraorbitals are visible in MCSN 8072 and 5827 (Figs. 7 and 9). The posterior anterior infraorbital is longer than deep, and its lateral surface is traversed by a longitudinal groove and, although not well preserved, numerous radiating grooves, as is the case of the first two infraorbital bones. The antorbitals are very slender, crescent-shaped tubular bones with slightly larger dorsal and smaller anterior arms (MCSN 5827; Fig. 9).

Between the infraorbitals and preopercle the cheek is covered with numerous suborbital bones. The most dorsal suborbital bone is the largest and occupies the whole length between the preopercle and the circumborbitals, with a depth of about 75% of its length. Ventral to this large suborbital, the other suborbitals form a mosaic of numerous and very small bones extending up to a level at the centre of the orbit. The pattern of suborbitals is variable and in several specimens the second and sometimes the third suborbital are as long, though less than half the depth of the dorsalmost suborbital (MCSN 5827, 8009, 8083; Figs. 9 and 12). All of the suborbital bones are ornamented.

The preopercle is very slender and crescent-shaped, but only slightly curved anteriorly, and oriented almost vertically. A series of pores close to the posterior border of the bone indicate the exits of tubules branching from the preopercular sensory canal. These tubules are clearly visible in the largest specimens (e.g., MCSN 8448). The posterior suborbital bones overlap the anterior margin of the preopercle and the dorsal portion of the preopercle is almost completely hidden by the suborbitals in the largest specimens.

Opercular series and branchiostegal rays

The opercle is 10–30% deeper than it is long, with almost straight dorsal and anterior borders. The ventral border is only very slightly convex, the posterior border is convexly curved, so that the bone is longest at the middle or at its ventral margin. On the medial surface there is a relatively large ovoid articular facet for the hyomandibula (PIMUZ T 486; Fig. 11A). The subopercle is smaller than the opercle, with a maximal depth of 35–40% of the maximal depth of the opercle. The dorsal border is only slightly concave and the ventral border is convex, so that the bone is roughly sickle shaped. It has a well-developed ascending process, which is tapering dorsally, but rather low; its height is about 20% of the maximal length of the subopercle and also c. 20% of the height of the anterior border of the opercle (MCSN 8469, 8001; Figs. 10 and 14). The anterior border of the subopercle is oblique, anteroventrally inclined. Both the opercle and subopercle are densely ornamented with ganoin tubercles and small ridges or patches. The interopercle is triangular, tapering rostrally, with a gently convex ventral border and straight dorsal and posterior borders. The maximal length of the interopercle is 1.2–1.5 times its maximal depth. Only the posterior margin of the interopercle is sparsely ornamented.

Medioventral to the subopercle there is the largest branchiostegal ray, which is distinguished as a branchiopercle because of its notably larger size and much stronger ornamentation than the remaining branchiostegals. Also, differing from the other branchiostegals and resembling the branchiopercle of Amia, this bone in T. longaeva has a broad and flatly truncated anterior end (MCSN 8009, 8469; Figs. 10 and 12; Hubbs, 1919; Grande & Bemis, 1998). The dorsal margin of the branchiopercle is overlapped by the subopercle and the ventral margin is densely ornamented in the same way as the opercular bones. The branchiopercle is followed by six branchiostegal rays, which decrease in size anteriorly and are not or only very slightly ornamented. A gular plate is absent.

Hyomandibula and hyoid arch

The hyomandibula is partially exposed in lateral view in several specimens (MCSN 8001, 8008, 8317, 8351; Fig. 14, Appendix S1: Figs. 2–4) and completely in medial view in PIMUZ T 486 (Fig. 11). The dorsal portion, representing about a third of the bone, is almost quadrangular with a long and oblique articular facet and straight vertical posterior border. There is a large and pronounced opercular process, which is very strong and broadens towards the opercle. Ventrally, the opercular process is connected with the vertical shaft through a thin lamina of bone. The vertical shaft is well ossified and broadens ventrally. The ventral border forms a posteroventrally directed facet probably for the articulation with a medial subopercular process, which is a cartilage that links the hyomandibula to the medial surface of the subopercle in living gars (Grande, 2010). The hyomandibular trunk enters the bone medially at the level of the opercular process and exits through a large opening on the lateral surface, approximately at the centre of the bone (MCSN 8317, PIMUZ T 486; Fig. 11, Appendix S1: Fig. 2).

A small symplectic is partially exposed and articulated in MCSN 8073 and PIMUZ T 4987 (Fig. 13A, Appendix S1: Fig. 6) and disarticulated and well exposed in MCSN 8009 (Figs. 12 and 13B). Based on these two fossils, we interpret the condition in T. longaeva as similar to the mode of articulation of the symplectic in Pachycormus curtus described by Patterson (1973: 274, Fig. 18; Fig. 13C, see below). Although there is no quadratojugal in this later taxon there is also a step on the medial surface of the quadrate, towards which the symplectic is attached (partially fused to the quadrate in the specimen NHMUK 32434 studied by Patterson, though he considered this might not be the typical condition).

There is a single, but large and approximately quadrangular hypohyal on each side (best preserved in MCSN 8008 and 8469; Fig. 10, Appendix S1: Fig. 4). The anterior ceratohyals are rather massive and well preserved in several specimens (MCSN 8001, 8008, 8009, 8351, 8469, PIMUZ T 486; Figs. 10–12 and 14, Appendix S1: Figs. 3–4). Each of these bones is subrectangular with strengthen and rather straight ventral border and gently curved antero- and posterodorsal corners. The bone is 2–2.5 times longer than it is deep. The posterior ceratohyal is only partially exposed and its shape cannot be reconstructed.

Palatoquadrate

The palatoquadrate is fully ossified, but it is only partially exposed and the individual bones cannot be described in detail (MCSN 8009, 8011, 8073, 8083, 8351; Fig. 12, Appendix S1: Figs. 3 and 7). The pars autopalatina is ossified. Dermo- and ectopterygoids bear the most powerful teeth, which are conical. The ectopterygoid is only exposed in the lateral view as a crescent-shaped bone with tapering anterior and posterior ends and only two teeth in the most anterior portion. The posteroventral portion of the ectopterygoid is articulating with the quadrate caudally. The quadrate is relatively small. The quadratojugal is also small and in some specimens it is tightly bound or partially fused (MCSN 8009, 8073) to the posteroventral border of the quadrate, but it is completely fused to the quadrate in PIMUZ T 4987 (Figs. 12 and 13A–13C, Appendix S1: Fig. 7). Immediately above the condyle, the inner surface of the quadrate forms a posterodorsally directed rounded shelf (Fig. 13B). Based on the conditions observed on the specimens MCSN 8009, 8073 and PIMUZ T 4987, we think that the anterior strengthened portion of the symplectic articulated and was able to slide on this inner shelf of the quadrate (Fig. 13C). The well-defined condyle for the articulation with the lower jaw is formed by the quadrate and quadratojugal together (Figs. 13A–13B).

Jaws

The two premaxillae are disarticulated and displaced in MCSN 8007, 8009 (Figs. 8, 12 and 13D–13E). The nasal processes are relatively short, with a depth less than two times the width of the toothed portion of the bone. The passage of the olfactory nerve is indicated by a broad groove and an open fenestra at the posteromedial portion of the nasal process. None of the specimens shows a complete olfactory foramen and the right premaxilla in MCSN 8007 seems to be completely preserved, so we think that the nasal process does not form a complete olfactory foramen in T. longaeva (Fig. 13D). A foramen at the base of the nasal process probably corresponds to the passage of the palatine ramus of the facial nerve (Allis, 1898; Fig. 13D). The left premaxilla is very nicely exposed in medial view in MCSN 8009 showing a broad medial “ascending” process at the anteromedial corner of the bone (Fig. 13E). This “ascending” process is much shorter than the nasal process and has a very irregular border. The premaxilla has 6 pencil-like teeth, which are notably smaller than the teeth on the dentary, but larger than the ones on the maxilla.

The maxillae have a strong and very long rod-like anterior articular process and a deep supramaxillary notch (MCSN 5827, 8001, 8009, 8072; Figs. 7, 9, 12 and 14). The maxillary paddle is between 3 and 3.5 times longer than deep and covers the anterior half of the coronoid process of the lower jaw laterally. The articular process of the maxilla is very long and rod-like. The dorsal border of the maxilla elevates very rapidly posterior to the articular process and turns abruptly downwards producing a deep supramaxillary notch. The single supramaxilla accommodates along the dorsal border behind the supramaxillary notch. The two bones together make an almost perfectly paddle shaped element. Small conical maxillary teeth are preserved in a few specimens (MCSN 8001, 8009, 8072, 8073; Figs. 7, 12 and 14, Appendix S1: Fig. 7), but their total number and distribution are unknown. The maxillary teeth are notably smaller than the dentary teeth and the smallest marginal teeth in the mouth (Fig. 14).

The lower jaw is formed by the dentary, angular, surangular, retroarticular, articular, prearticular and coronoid bones. The dentary has a long toothed anterior portion and forms well-developed coronoid and posteroventral processes (MCSN 8009, 8072, 8351; Figs. 7 and 12, Appendix S1: Fig. 3). There are 9–10 dentary pencil-like teeth, which are the largest marginal teeth on the jaws. The height coronoid process is 40–45% of the total length of the lower jaw. Above the posteroventral process, the posterior border of the dentary is almost perpendicular to the major longitudinal axes of the bone. Along the toothed portion of the dentary, the dorsal margin forms a medial shelf on which the coronoids are attached (MCSN 8008, 8001, 8413; Fig. 14, Appendix S1: Fig. 4). There are two coronoid bones bearing strong pencil-like teeth similar in shape to those of the dentary. There are two rows of teeth on the coronoids. The teeth on the lateral row are larger, approximately of the same size as the dentary teeth, but the teeth on the medial row are small. The prearticular is located on the medial surface of the dentary posterior to the coronoids (MCSN 8001; Fig. 14). The prearticular teeth are roughly arranged in two rows and are of similar size than the lateral coronoid teeth. The posteroventral process does not reach the posterior border of the lower jaw and the posterior portion of the ventral border of the lower jaw is formed by the retroarticular. The retroarticular is very large and it is exposed in several specimens (MCSN 8072, 8083, 8469; Figs. 7 and 10). It is sutured to the angular and forms the ventral border of the lower jaw caudal to the posteroventral process of the dentary. The surangular is relatively large, articulating dorsal to the angular and posterior to the dentary, and the dorsal border is convex. The concave articular facet is directed posterodorsally and it is formed by the articular alone.

Girdles and paired fins

The pectoral girdle is formed by the posttemporal, supracleithrum, presupracleithrum, cleithrum, and four postcleithra. The posttemporal is approximately triangular resembling the large extrascapular, but it is larger and anteriorly overlapped by this bone. Therefore, the exposed portion of the posttemporal is extremely similar in size and shape to the extrascapular. Also like the extrascapulars, the two posttemporal bones meet at the dorsal midline. Each posttemporal has a rounded ventral process for the articulation with the supracleithrum (MCSN 8001, 8009, 8072, 8073, PIMUZ T 486, 2848, 4785; Figs. 7, 11–12 and 14A).

The supracleithrum is only completely exposed in PIMUZ T 486, where it is preserved in medial view (Fig. 11). There are no medial dorsal or ventral processes as in the gars (Grande, 2010). The bone narrows ventrally and is broadest at its dorsal end, where it forms a concave articular facet directed dorsally, which receives the rounded process of the posttemporal forming a ball-and-socket articulation (Wiley, 1976). Directly anterior to this facet, a small ovoid bone, which is here named subposttemporal bone, places between the flattened anterodorsal border of the supracleithrum and a concave facet in the medial ventral margin of the posttemporal. A subposttemporal bone has been otherwise only observed in the callipurbeckiid semionotiform Macrosemimimus fegerti (A López-Arbarello, pers. obs., 2011). In the latter species this bone sits directly on the concave articular facet of the cleithrum indicating that it is an aid to the ball and socket articulation. The lateral line enters the supracleithrum at the middle of the bone and obliquely runs to exit through the dorsal border entering the posttemporal for a brief trajectory through the anteroventral portion of this bone. The lateral surface of the posterodorsal margin of the supracleithrum is variably ornamented with short ridges and tubercles. A small presupracleithrum is preserved in several specimens (MCSN 8009, 8072, 8073, PIMUZ T 486, 4790, 4987; Figs. 7 and 11–12, Appendix S1: Figs. 6–7).

The cleithrum is crescent-shaped in lateral view and has a single row of denticles lining the limit between the lateral and branchial surfaces of the bone (MCSN 8072, 8073, PIMUZ T 486; Figs. 7 and 11, Appendix S1: Fig. 7). The branchial portion of the cleithrum consists of a very broad lamina of bone (PIMUZ T 8001, 8009; Figs. 12 and 14A), which is probably homologous to the medially directed lamina of the cleithrum in Boreosomus (Nielsen, 1942) or the medial wing of the cleithrum in gars (Wiley, 1976; Grande, 2010). Posterior to the cleithrum there are four postcleithra, three of which are tightly articulated and form together a half crescent shape (MCSN 8072, 8073; Fig. 7; Appendix S1: Fig. 7). The dorsalmost postcleithrum is the largest and tapers dorsally. The second postcleithrum is almost rectangular and the third and much smaller postcleithrum is oval shaped. The fourth postcleithrum is also oval and a little smaller than the third postcleithrum. This most anterior postcleithrum is detached from the others and it is usually not preserved.

The endochondral elements of the pectoral girdle are only partially exposed, but there are at least five radials (MCSN 8001, 8007, 8072, 8073; Figs. 7–8 and 14A, Appendix S1: Fig. 7). The pectoral fin web is made up of 17–18 lepidotrichia (MCSN 8009, 8073; Fig. 14A, Appendix S1: Fig. 7). There are numerous fringing fulcra along the first pectoral fin ray, which have the typical ganoin ‘nail’ that characterizes the genus. The fringing fulcra are fused to each other towards the base of the fin, but they are not fused to the base of the first pectoral ray. The shape of the base of the first pectoral fin ray is variable between individuals and probably represents ontogenetic variation (see below).

The basipterygia are exposed in MCSN 8001 and 8475 (Appendix S1: Fig. 1). They are elongated and slender bones almost equally expanded proximally and distally. The pelvic fins are always imperfectly preserved so that the total number of rays is uncertain but seven rays are preserved on the right pelvic fin of MCSN 4974, 8072, and 8475 and this is most probably the complete set of pelvic rays. A series of fringing fulcra garnished with ganoin ‘nails’ lay on the first pelvic ray.

Dorsal and anal fins

The dorsal fin web is made of 14–16 rays extending backwards a little behind the vertical level at the end of the anal fin (Fig. 15A; Appendix S1: Figs. 4A and 7). There are a variable number of small basal fulcra at the origin of the dorsal fin, which are usually fused to each other at their bases and carry the first fringing fulcrum, and a series of small though garnished with ganoin ’nails’ fringing fulcra lay along the first dorsal fin ray. The anal fin is relatively small consisting of six rays (Fig. 15B). There are a few small basal fulcra at the origin of the fin and a series of fringing fulcra on the leading edge of the fin. The first ray in both dorsal and anal fins is unbranched and much shorter than the second ray.

Caudal fin and body lobe

The caudal fin is abbreviated heterocercal with a well-developed body lobe including 8 to 9 inverted rows of scales and a median dorsal scute (Figs. 12 and 15C; Appendix S1: Figs. 3 and 6–7). There is a marginal row of seven scales that merges with the adjacent inverted row distally. This second inverted row is the longest, made up of 11 to 14 scales, varying in the number of the most distal scales forming the sharp tip of the body lobe.

The caudal fin web is forked with equally long ventral and dorsal lobes. There is a variable number (1–5) of small paired and unsegmented basal fulcra followed by 3–5 short segmented rays which are bearing fringing fulcra. The first principal ray is the first ray, which is unbranched, but as long as the following ray and bears fringing fulcra. There are 11 principal caudal fin rays in the ventral lobe below the lateral line, and 11 to 12 rays in the dorsal lobe. Except for one or two rays placed in the central part of the fin, the basal segment of the dorsal principal rays are very long and located oblique to main axis of the ray, approximately following the orientation of the body lobe scale rows (MCSN 8475; Fig. 14C). The basal segment of the dorsal principal rays of the body lobe have small dorsal processes (MCSN 8009, 8073, 8413, PIMUZ T 4987; Fig. 12; Appendix S1: Fig. 6–7). The caudal fin rays are laterally covered with patches of ganoin.

Body squamation

The body of Ticinolepis longaeva is covered with numerous and relatively small thick rhomboid scales. There are 39–40 scales along the lateral line and between 14 and 22 scales above the lateral line along the vertical row immediately anterior to the dorsal fin (intraspecific variation, see below). In the most anterior portion of the flank, in the first three or four vertical rows, the scales are approximately quadrangular, but they are shallow, longer than deep in the rest of the body. As usual, the scales are gradually smaller starting from the most anterior scales of the lateral line in dorsal, ventral and posterior directions, but in Ticinolepis the ventrum is covered with extremely shallow scales (Figs. 5 and 11–12; Appendix S1: Figs. 3–4 and 6–7). This feature is especially noticeable in this species.

The scales are rhomboid, with smooth surface and variably serrated posterior border, that is to say that scales with serrated borders are found in some, but not all of the specimens and in some regions, but not all over the body. The distribution of these serrated scales is capricious and we have not been able to identify any particular pattern. The scales have the usual peg-and-socket articulation and a broad smooth anterior surface devoid of ganoin for the articulation with the adjacent scales, but there are no distinct longitudinal processes.

There is no distinct dorsal ridge of scales and only one specimen has distinct, though small preanal and precaudal scutes (MCSN 8073; Fig. 15B; Appendix S1: Fig. 7). All the fins are preceded by one or a few small basal fulcra and fringed with numerous small fringing fulcra, which are distinctly ornamented with a thick oval patch of ganoin.

Ticinolepis crassidens sp. nov.	
urn:lsid:zoobank.org:act:87791836-F508-4D3A-97DD-160A32751F01	
(Figs. 5B, 6B, 6D and 16–22; Appendix S1: Figs. 9–11)	

p v. 1998 Archaeosemionotus sp. Bürgin: Fig. 8H, Table p. 7

p v. 1999a Archaeosemionotus sp. Bürgin: Appendix 1

Holotype. PIMUZ T 273 (Fig. 5B).

Paratypes. PIMUZ T 237, 240, 248, 256, 257, 265, 277, 278, 315, 331, 341, 358, 401, 413, 438, 638, and 4350.

Type locality. P. 902/Mirigioli, Monte San Giorgio (Meride, Canton Ticino, Switzerland; Fig. 1).

Type horizon. Upper Besano Formation, early Ladinian, lowermost E. curionii Ammonoid Zone.

Etymology. The species name crassidens derives from the latin crassus, thick, and dens, tooth, referring to the strongly tritoral dentition of this species.

Referred specimens. PIMUZ T 2823 and 2835 from the uppermost Besano Formation (Val Porina mine).

Diagnosis. Species of Ticinolepis differing from the other species of this genus in the following combination of characters: tritoral teeth on dermopalatin, ectopterygoid, coronoids and dentary; lower jaw and skull roof bones totally or partially fused; 6 infraorbital bones; 8–9 principal caudal fin rays below the lateral line; only 8–9 flank scales above lateral line; marginal row of body lobe with 9 scales; second and third inverted rows of body lobe equally long, merging distally, made up of 9–10 scales in each row and two common scales forming the tip of the body lobe.

Figure 16 Ticinolepis crassidens sp. nov., braincase in PIMUZ T 273 (holotype).

(A) Photograph. (B) Line drawing. The basipterygoid process of the parasphenoid is indicated with a red circle. Scale bars = 5 mm.

Figure 17 Ticinolepis crassidens sp. nov., skull of PIMUZ T 2835 preserved in dorsal (skull roof) and left lateral views.

(A) Photograph. (B) Line drawing. Scale bars = 5 mm.

Figure 18 Ticinolepis crassidens sp. nov., PIMUZ T 256 preserved in right dorsolateral view.

(A) Overview of the complete specimen; note two specimens of small high-spired gastropods (?Coelostylinidae), preserved in the same layer of dolomite (det. V. Pieroni, 08.03.2016); scale bar = 2 cm. (B) Skull; scale bar = 5 mm. (C) Caudal fin; scale bar = 5 mm.

Figure 19 Ticinolepis crassidens sp. nov., PIMUZ T 4350 preserved in left lateral view.

(A) Photograph of the anterodorsal portion of the skull showing the nasals and circumborbital bones; scale bar = 5 mm. (B) Line drawing of the skull; the hyomandibula and the rounded ventral process of the posttemporal are indicated with blue and orange circles, respectively; scale bar = 5 mm. (C–D) left palatoquadrate partially preserved in median view; scale bars = 5 mm. (C) Photograph. (D) Line drawing. (E) Overview of the complete specimen; arrow points to the disarticulated and displaced left palatoquadrate; scale bar = 2 cm. (E) Caudal peduncle and fin; scale bar = 5 mm.

Figure 20 Ticinolepis crassidens sp. nov., skull in PIMUZ T 277 preserved in right lateral view.

Scale bar = 5 mm.

Figure 21 Ticinolepis crassidens sp. nov., photographs of PIMUZ T 413 preserved in right lateral view.

(A) Overview of the complete specimen; scale bar = 2 cm. (B) Skull; scale bar = 1 cm. (C) Caudal fin; scale bar = 1 cm.

Figure 22 Ticinolepis crassidens sp. nov., skull and dentition in PIMUZ T 273 preserved in left lateral view.

(A) Photograph of the skull; scale bar = 5 mm. (B) Line drawing of the skull; the basipterygoid process of the parasphenoid is indicated with a red circle; scale bar = 5 mm. (C) Line drawing of the vomerine and dermopalatine dentition; scale bar = 2 mm. (D) Line drawing of the dentition of the lower jaw; scale bar = 2 mm.

Pterygial formula:

D17–19P6–7A15–16C26–28T31–34.

Anatomical description

Ticinolepis crassidens is smaller than the type species of the genus, reaching up to 11 cm SL (Fig. 5). The general shape of the body is fusiform with a relatively large head representing approximately 33% of the SL. Generally, the material of T. crassidens is more poorly preserved than the specimens representing the type species, in particular the material from the Meride Limestone. In the material of T. crassidens the skull bones are mostly disarticulated and usually broken. Therefore, it is not possible to take accurate morphometric measurements and the anatomical information is not as complete as for the type species of the genus.

Braincase, parasphenoid and vomers

The braincase is very well preserved and partially exposed in left lateral view in PIMUZ T 273 (Fig. 16). The exposed portion is remarkably similar to the braincase of the specimen NHMUK 37795a of Ionoscopus cyprinoides (Maisey, 1999), except that sutures are not discernible and all the bones seem to be fused, including the large and high ascending process of the parasphenoid (= processus ascendens posterior; see ‘Discussion’), which is almost perpendicular to the parasphenoid bar, being only slightly inclined posteriorly. It is tightly bound to the sphenotic dorsally and fused to the prootic posteriorly. At the base of the ascending process there is an anteriorly directed triangular process that resembles the pterosphenoid process of the parasphenoid of Amia (indicated with a red dot in Figs. 16 and 22; Grande & Bemis, 1998), but it is not sutured to the pterosphenoid. Similar free processes, not articulated with other bones, are present in the ionoscopiforms Ionosocpus cyprinoides and Oshunia brevis (Maisey, 1999). We interpret this process as the processus ascendens anterior, which corresponds to the basipterygoid process (see ‘Discussion’ below). Ventral to this process there is a notch related to the passage of the efferent pseudobranchial artery (Patterson, 1975: 531) or the posterior palatine branch of the cranial nerve VII (Grande & Bemis, 1998), or both (Allis, 1897; Bjerring, 1972; Bjerring, 1977; Jarvik, 1980). Posteriorly, at the base of the ascending process, there is a notch corresponding to the passage of the internal carotid artery (Allis, 1897; Patterson, 1975; Grande & Bemis, 1998) and probably also the pharyngeal brunch of the glossopharyngeal nerve (Bjerring, 1972; Bjerring, 1977; Jarvik, 1980). The lateral wall of the otic region is completely ossified embracing the areas corresponding to the prootic, opisthotic and pterotic. In the prootic area there is a large foramen corresponding to the passage of the hyomandibular trunk and the jugular vein, and more dorsally a smaller foramen corresponding to the spiracular canal (Allis, 1897; Bjerring, 1972; Bjerring, 1977; Patterson, 1975; Jarvik, 1980; Grande & Bemis, 1998). The opening of the spiracular canal is placed immediately anteroventral to the hyomandibular facet, which is completely ossified between the areas corresponding to the prootic, pterotic and sphenotic. The facet is anteroventrally inclined in approximately 35°respect to the longitudinal axis of the parasphenoid + basioccipital. In the occipital region, the exo- and basioccipitals are coossified. Immediately posterodorsal to the vagus foramen there is a cranio-spinal process, which is homologous to the intercalar in basal teleosts according to Patterson (1975). The dorsal portion of the occipital region is not well visible and, thus, an epioccipital is not discernible.

In the orbital region, anterior to the ascending process of the parasphenoid the pterosphenoid is visible, but no details can be described. The pyramidal sphenotic is tightly bound to the prootic and the ascending process of the parasphenoid. Some specimens have badly preserved remains of the orbitosphenoid, which was large, reaching the parasphenoid ventrally (PIMUZ T 2823). The ethmoidal region is strongly ossified, with large lateral ethmoids. The vomers are not exposed, but the strongly tritoral vomerine dentition (Fig. 16) indicates that the bones were probably fused.

Skull roof and snout

The frontals, parietals and dermopterotics are tightly ankylosed or partially fused and the sutures are almost indistinguishable (best preserved in PIMUZ T 438, 2835; Fig. 17). Therefore, it is not possible to describe the shape and proportions of the individual bones. As a whole, the skull roof is c. 35% broader posteriorly than anteriorly. There is an approximately rectangular interorbital constriction and the anterior portions of the frontals make rounded lateral antorbital processes. The middle and posterior pit lines are indicated by grooves at the rear of the skull roof. The middle pit lines run through the dermopterotics and parietals, the posterior pit lines are indicated by grooves almost parallel to the posterior margin of the parietals. The dermopterotics project forward forming distinct anterolateral processes (PIMUZ T 438; Appendix S1: Fig. 8). Resembling the type species of the genus, there is a single pair of extrascapulars in T. crassidens. They are large and triangular, articulating with the dermopterotics and parietals and reaching the dorsal midline (PIMUZ T 256, 257, 277, 401, 638; Fig. 18, Appendix S1: Fig. 10).

The nasals are less strongly ossified and narrower than in the type species (PIMUZ T 413, 2823 and 2835; Figs. 17 and 19). They are neither ornamented nor tightly articulated with the frontals, the rostral or between them, so they were most probably imbedded in the skin, apparently only attaching to the longitudinal median ridge made by the large nasal processes of the premaxillae. They are thin laminar bones and their margins are generally broken and incompletely preserved, except in PIMUZ T 4350 (Fig. 19), where they are exceptionally well preserved. Each nasal is broadest posteriorly, with slightly laterally curved anterior end and, thus, the lateral margin is deeply concave in the anterior portion, probably surrounding the anterior external nares. Longitudinally, each nasal is traversed by a tube carrying the supraorbital sensory canal, which turns laterally close to the anterior margin of the bone. The rostral is roughly rhomboidal with two lateral horns including a tube for the ethmoidal commissure and a medial caudally directed knoll producing a roughly V-shaped dorsal border (PIMUZ T 277; Fig. 20).

Circumborbital series and cheekbones

The dermosphenotic and the series of supraorbitals are most completely preserved in PIMUZ T 4350 (Fig. 19). Each dermosphenotic is small, usually rectangular, but it might have an anteriorly directed arm extending above the orbit. The posterodorsal portion of the bone fits into the ‘notch’ formed by the lateral border of the frontal and the anterior border of the anterolateral process of the dermopterotic (PIMUZ T 438). This area is also probably sitting on the lateral process of the sphenotic. Along the dorsal rim of the orbit, the dermosphenotic is followed by four small supraorbitals, which are gradually larger in anterior direction. The supraorbitals are weakly ornamented on their external surfaces, but the whole supraorbital series produces a longitudinal ridge along the internal surfaces of the bones, probably for attachment to the rim of the frontal.

Ventral to the dermosphenotic, the circumborbital series is completed with a series of 6 infraorbital and at least two anterior infraorbital bones (PIMUZ T 277, 315; Fig. 20). The anterior infraorbitals are very delicate and thin laminar bones, and are thus only rarely preserved. The anterior infraorbitals are approximately rectangular in shape, a little longer than deep and gradually shallower in anterior direction. The first infraorbital, placed at the anterior rim of the orbit, does not contact the anterior supraorbital and, thus, the orbit is open anteriorly. The first infraorbital is small and has no particularly distinct features. As in the type species of Ticinolepis gen nov., the second infraorbital is the largest in the series, but differing from this other species, the bone in T. crassidens is not fan-shape but approximately rectangular and similar to the other infraorbitals. The infraorbitals three to six complete the posteroventral and posterior rim of the orbit. Their relative size and shape are somewhat variable from rectangular to tubular between specimens, but the infraorbital immediately adjacent to the dermosphenotic is always very small (best exposed in PIMUZ T 277 and 4350; Figs. 19–20). The antorbitals are small crescent shaped bones, with the anterior arm longer than the depth of the dorsal arm (PIMUZ T 277).

As in the other species of the genus, there is a mosaic of small suborbitals covering the cheek posteroventral and posterior to the circumborbital series in T. crassidens. Also as in T. longaeva, the dorsalmost suborbital is notably larger than the other suborbitals, covering the area between the infraorbitals and the preopercle approximately up to the level of the parasphenoid. This largest suborbital is also about 30% longer than it is deep. The number and arrangement of the other suborbital bones is variable between individuals, but the whole mosaic of suborbitals completely covers the palatoquadrate laterally, reaching up to the level of the coronoid process of the lower jaw, at the centre of the orbit (PIMUZ T 277; Fig. 20).

All suborbitals and circumborbital bones are densely ornamented with small, adjacent ganoin patches. In the dermosphenotic and infraorbitals the ornamentation is not complete and the orbital margin of the bones is smooth. Similarly, only the posterodorsal margin of the most anteroventral suborbitals is ornamented (PIMUZ T 277, 331, 2823).

The preopercle is very slender and crescent-shaped (best preserved in PIMUZ T 277, 331 and 4350; Figs. 19–20). The anterior margin is overlapped by the suborbitals and the preopercular sensory canal is indicated by a series of pores at the posterior border of the bone.

Opercular series and branchiostegal rays

The opercle is large and about 50% deeper than long and has almost straight rostral and ventral borders and gently convex dorsal and caudal borders (Figs. 18 and 21). It is shortest at the dorsal margin and longest at about two thirds of its depth. The subopercle is relatively large and sickle-like shape with a maximal depth (excluding the anterodorsal process) of about 30% of the depth of the opercle. The ascending process is robust, with long base, reaching up to a level at about half of the depth of the opercle. The anterior border of the subopercle is oblique, slightly inclined anteroventrally. Both the opercle and subopercle are densely ornamented with small, adjacent patches covered with ganoin (PIMUZ T 257, 277, 413, 2823). The interopercle is triangular, rather large, but relatively short, with a maximal length only about 20% of its maximal depth, and with a blunt rostral tip (PIMUZ T 413; Fig. 21). Only the posterior margin of the interopercle is ornamented in the same way as the opercle and subopercle (PIMUZ T 257, 277, 413, 2823). There is no evidence of a median gular. A branchiopercle is preserved in PIMUZ T 315 and 413. The complete set of six branchiostegal rays is only preserved in PIMUZ T 315.

Hyomandibula and hyoid arch

The hyomandibula is partially exposed in lateral view in PIMUZ T 273 and 4350 (Figs. 16 and 19). In the later it is partially covered by suborbitals and the preopercle, but in the first of these specimens the dorsal portion of the hyomandibula is well exposed. It is well ossified, rather slender and trapezoid. There is a broad and blunt opercular process. The ventral shaft is almost vertical, slightly inclined posteroventrally, and rather slender. There are two dorsoventrally directed parallel grooves ending in small foramina on the lateral surface of the hyomandibular shaft. These two grooves represent the curse of the hyomandibular trunk, which includes three main branches in Amia. The two grooves suggest that the r. mandibularis facialis and the r. mandibularis lateralis separated from the posterior r. hyoideus bevore exiting the lateral surface of the hyomandibula (see description of the curse of these branches in Jarvik, 1980: 51). Better preserved material is needed to confirm this interpretation.

There is a robust quadrangular hypohyal visible in PIMUZ T 315 and 4350 (Fig. 19B). The anterior ceratohyal is well visible in several specimens (PIMUZ T 277, 315, 438, 2823, and 4350; Figs. 19–20, Appendix S1: Figs. 8 and 11) and is roughly rectangular with no median constriction. The bone is less than two times longer than deep and, thus, it is relatively shorter than in the other species of Ticinolepis. The posterior ceratohyal is only partially exposed in PIMUZ T 277 and 2823.

Palatoquadrate

The palatoquadrate is fully ossified, but it is not possible to delimit the individual bones. Although an autopalatine bone cannot be delimited, the pars autopalatina is clearly ossified (PIMUZ T 273; Fig. 16). At least one dermopalatine and the ectopterygoid bear powerful molariform teeth (PIMUZ T 4350; Figs. 19C–19D). These bones are relatively large and powerful. The quadrate has a distinct massive irregularly shaped body and a well-defined condyle. There is a splint-like quadratojugal firmly attached to the posteroventral border of the quadrate.

Jaws

The two premaxillae are incompletely exposed in the studied specimens and only the toothed portion is visible. The nasal processes are hidden and, thus, their length is unknown.

The maxillae and supramaxillae are very poorly preserved. Based on the few specimens, in which these bones are almost completely preserved (PIMUZ T 315, 341, 2823; Appendix S1: Fig. 11), they are very similar to the maxillae and supramaxillae of T. longaeva, with approximately the same proportions and similar shape variation.

The lower jaw is always exposed in lateral view and the bones are all fused together forming a massive jaw with a very strong and high coronoid process (Figs. 17–18 and 20–22; Appendix S1: Figs. 8 and 11). The maximal depth of the bone at the coronoid process is c. 60% of the maximal length of the jaw. The symphysis is relatively shallow with a depth of about 20% of the maximal length of the jaw. The almost horizontal toothed portion of the dentary is c. 55% of the maximal length of the jaw. The anterior edge of the coronoid process is concave (PIMUZ T 273, 277).

Except for the maxillary teeth, the dentition is very strong (Figs. 17–22; Appendix S1: Figs. 8 and 11). Each premaxilla bears 4–5 teeth (Appendix S1: Fig.9) and there is a marginal row of 7–8 teeth on the dentary (Fig. 21D). Two prearticular teeth are laterally exposed posterior to the dentary teeth in PIMUZ T 273 and 277 (Figs. 20 and 21D). The coronoid bones are only partially exposed and are never found detached from the jaw and, thus, they were probably fused to the dentary. Large molariform coronoid teeth are well exposed in most of the specimens. These and the vomerine teeth are the largest teeth in this species (Figs. 21E–21D).

Girdles and paired fins

Like in the type species, the pectoral girdle of T. crassidens includes a posttemporal, a supracleithrum, a small presupracleithrum, a cleithrum, and an unknown number of postcleithra (Figs. 17–18; Appendix S1: Figs. 9–11). The posttemporal is also large, subtriangular, and reaches the dorsal midline. As in the type species, the posttemporal of T. crassidens presents a rounded process for articulation with the supracleithrum (indicated with an orange dot in Fig. 19B). The posttemporal bears along its ventral margin a short streak of the sensory canal, which represents the connection of the lateral line with the cephalic sensory canal system. A further part of this connection pierces the supracleithrum obliquely (Appendix S1: Fig. 9). This latter element has a deep, almost rectangular shape, with ventral part tapering into a slightly pointed tip. A small notch at the caudal margin of the supracelithrum marks the entering of the lateral line. The presupracleithrum is exposed in PIMUZ T 256 and 2835 (Figs. 17–18). The large cleithrum is only partially preserved and shows a distinct medial ridge with denticles running along its lateral surface. There is a shallow incurvation on the caudal edge, just behind the pointed tip. The posttemporal and presupracleithrum are fully and the supracleithrum is partially ornamented by irregular ridges and tubercles, whereas no ornamentation is found on the external surface of the cleithrum. The pectoral fin is composed of 13–16 segmented and, with the exception of the first, distally branched fin rays (Appendix S1: Fig. 11). The leading edge of the fin is composed of a spinous basal fulcrum and a paired series of well-developed fringing fulcra.

The pelvic bones are not exposed or preserved. The small pelvic fins consist of four to five distally segmented and branched rays. A single spinous basal fulcrum precedes the leading edge of the fin, which bears a series of fringing fulcra.

Dorsal and anal fins

The dorsal and anal fins are very poorly preserved in the material of T. crassidens (Figs. 5B, 18 and 21; Appendix S1: Fig. 8). They are generally similarly built as in the other species. The dorsal fin includes 13 rays in PIMUZ T 256 and there are eight anal fin rays in PIMUZ T 237.

Caudal fin and body lobe

The caudal fin is abbreviated heterocercal with a body lobe including 8–9 inverted rows of scales and a median dorsal scute. Differing from the other species, in T. crassidens the marginal row includes nine scales and the second and third inverted rows are equally long, merging distally, made up of 9 (PIMUZ T 256) to 11 (PIMUZ T 413) scales in each row and two common scales forming the tip of the body lobe (Figs. 17C, 18F and 20C).

The caudal fin web is forked with equally long ventral and dorsal lobes. There three to four small unsegmented basal fulcra followed by three short segmented rays, which might bear fringing fulcra. The first principal ray is the first ray, which is unbranched, but as long as the following ray and bears fringing fulcra. There are only nine principal caudal fin rays in the ventral lobe, below the lateral line, and ten rays in the dorsal lobe.

Body squamation

Like in the type species, the body of T. crassidens is covered with thick rhomboid scales, which are generally quadrangular, slowly changing from deeper than long in the anterior vertical rows, to slightly longer than deep in the centre of the body and the caudal peduncle. As usual in Ticinolepis the ventrum is covered with extremely shallow scales, though in this species the longitudinal rows of shallow scales are not as numerous as in the type species (Figs. 5B, 18 and 21; Appendix S1: Fig. 8). The scales have smooth surfaces and smooth posterior border, except for the scales traversed by the lateral line, which might have a posterior excavation. The scales have the usual peg-and-socket articulation and a broad smooth anterior surface devoid of ganoin for the articulation with the adjacent scales, but there are no distinct longitudinal processes. There is no distinct dorsal ridge of scales and no preanal scute.

Discussion

Comparative anatomy

The anatomy of the two species of Ticinolepis is noteworthy because it includes a mixture of typically ginglymodian and halecomorph features. Already the external morphology is conflictive. The general aspect of these fishes, with a relatively large head, the body completely covered with rhomboid scales and abbreviated heterocercal caudal fin agrees with a generalized basal neopterygian. The presence of anterior infraorbital bones and a mosaic of suborbital bones is typical of Ginglymodi, but the large nasal bones covering the nasal capsule completely, the presence of a branchiopercle and a supramaxillary notch, and the series of very shallow scales covering the ventrum indicate affinities with the halecomorphs. These and other puzzling features are discussed in detail within this section.

The braincase in Ticinolepis resembles most closely the braincases of some basal halecomorphs like Caturus furcatus or Ionoscopus cyprinoides. The hyomandibular facet is completely ossified between the prootic, pterotic and sphenotic, as is the case in some basal halecomorphs (e.g., Caturus furcatus NHMUK 20578). In basal teleosts, parasemionotids and various fossil holosteans the facet is in an equivalent position, but it probably remained cartilaginous throughout life (e.g., “Pholidophorus” germanicus, Siemensichthys macrocephaluans, Pachycormus curtus, Heterolepidotus in Patterson, 1975). In Macrosemimimus toombsi (NHMUK P. 34511), Scheenstia mantelli (in Gardiner, 1960) and in the gars (Grande, 2010) the prootic is comparatively smaller and there is a relatively large cartilaginous area between the sphenotic, prootic, basioccipital, epioccipital and dermopterotic. Also differing from the condition in Ticinolepis gen. nov., in the gars the hyomandibula articulates on the ventral flange of the dermopterotic (Jollie, 1984; Grande, 2010). The ethmoidal ossifications are generally small or not developed in ginglymodians (López-Arbarello, 2012; Cavin, Deesri & Suteethorn, 2013), except for Araripelepidotes, in which the lateral ethmoids are well ossified (Wenz & Brito, 1996). Contrary, there are large and well ossified lateral ethmoids in Ticinolepis gen. nov., which is also the condition found in Caturus furcatus (NHMUK 20578), Ionoscopus cyprinoides, Oshunia brevis and Calamopleurus cylindricus (NHMUK 37795, AMNH 11840 and AMNH 11840, respectively, in Maisey, 1999).

The parasphenoid in Ticinolepis also resembles the parasphenoid of basal halecomorphs in having relatively large, vertically oriented ascending processes, which articulate with the sphenotic (e.g., Caturus furcatus NHMUK 20578; Ionoscopus cyprinoides NHMUK 37795, Oshunia brevis AMNH 11840 in Maisey, 1999). The ascending processes of the parasphenoid are well developed, but comparatively smaller in Macrosemimimus lennieri (NHMUK P. 34511) and Scheenstia mantelli (in Gardiner, 1960), and they are separated from the sphenotic. In the gars the parasphenoid lacks the ascending processes (Wiley, 1976; Grande, 2010).

Two ascending processes are identified in the parasphenoid of early actinopterygians: processus ascendens anterior and posterior (e.g., Stensiö, 1925; Stensiö, 1932; Aldinger, 1937; Nielsen, 1942; Bjerring, 1977; Jarvik, 1980; Štamberg & Zajíc, 2000). The processus ascendens posterior is more widely present among actinopterygians and it is usually referred to as the ascending process (e.g., Patterson, 1975; Patterson, 1982; Gardiner, 1984; Grande & Bemis, 1998; Coates, 1999; Choo, 2011; Giles et al., 2015). The processus ascendens anterior is related to the processus basypterigius of the endochondral neurocranium (Stensiö, 1932: 273; Nielsen, 1942: 105, 324), and it is thus usually referred to as the basypterigoid process of the parasphenoid (e.g., Gardiner, 1984), especially in neopterygians (e.g., Patterson, 1975; Grande & Bemis, 1998; Grande, 2010). The precessus ascendens anterior is poorly developed in several taxa including Pteronisculus and Boreosomus (Nielsen, 1942; Lehman, 1952), and Amia and other halecomorphs (Jarvik, 1980; Grande & Bemis, 1998—their pterosphenoid process of the parasphenoid), and it is absent in several actinopterygians including the chondrosteans Birgeria (Nielsen, 1949), Condorlepis (López-Arbarello, Sferco & Rauhut, 2013), Chondrosteus (Hilton & Forey, 2009), and Acipenser (Hilton, Grande & Bemis, 2011), but also in Australosomus (Nielsen, 1949) and Perleidus (Stensiö, 1932; Lehman, 1952). The two species of Ticinolepis differ in the degree in which the basipterygoid processes of the parasphenoid are developed. The basipteryogid processes are larger and more evident in the specimens of T. longaeva (Fig. 7) than in the only specimen of T. crassidens that shows this feature (PIMUZ T 273; Fig. 16). This difference is challenging because the condition shown by T. longaeva resembles the massive basipterygoid processes characteristic of several fossil ginglymodians and Jurassic teleosts, while the condition in T. crassidens resembles the case in halecomorphs.

In the skull roof, the two broad frontals of Ticinolepis cover the orbital and most of the temporal region, but they do not extend anterior to the orbit over the ethmodial region. This is the most common condition among halecomorphs (e.g., Grande & Bemis, 1998). In contrast, the frontals in ginglymodians and basal teleosts have well-developed antorbital portions (e.g., López-Arbarello, 2012; Arratia, 2013). Anterior to the frontals, the very large paired of nasals (with a dorsal surface area of more than 30% the dorsal surface area of the frontals), which are in contact at the midline, closely resemble the condition found in the amiiform genera Amia and Cyclurus (with dorsal surface areas of 15–20% of the dorsal surface area of the frntals; Grande & Bemis, 1998), which is clearly different from the much smaller and loosely attached nasals of ginglymodians or basal teleosts (e.g., López-Arbarello, 2012; Arratia, 2013). Although all these are only rough estimations, the nasals of ginglymodians are less than 10% the size of the frontals, even in taxa with comparably large nasals like Araripelepidotes or Pliodetes (A López-Arbarello, pers. obs., 2016).

The dermopterotics in Ticinolepis have well-developed descending laminae. The distribution of this feature is poorly investigated, but dermopterotic descending laminae have been described in amiiforms and other halecomorphs (Patterson, 1975; Grande & Bemis, 1998), in Semionotus elegans (Olsen & McCune, 1991), and in basal teleosts (Patterson, 1975). The dermopterotic of the gars lacks a descending lamina (Grande, 2010), but comparable laminae have been described for the supratemporo-intertemporal bone of some non-neopterygian actinopterygians (e.g., Birgeria mougeoti in Stensiö, 1921: 181; Acropholis stensioei in Aldinger, 1937: 41; Pteronisculus stensiöi in Nielsen, 1942: 120; Acipenser brevirostrum in Hilton, Grande & Bemis, 2011), so this feature might be plesiomorphic for neopterygians.

The circumorbital series in Ticinolepis includes anterior infraorbital bones, which as far as known are uniquely derived in Ginglymodi. Not only the presence of these bones, among crown neopterygians, the general shape and relative size of the supraorbital and infraorbital bones resemble most closely the condition in Semionotiformes (López-Arbarello, 2012). The mosaic of numerous and small suborbital bones is also a feature so far only known in ginglymodians though not in Semionotiformes; it is a relatively derived feature in Lepisosteiformes (López-Arbarello, 2012). Numerous suborbital bones might be present in ionoscopiforms (e.g., Robustichthys luopingensis Xu, Zhao & Coates, 2014), but in these fishes the suborbitals are not as small and numerous as in Ticinolepis, the lepisosteiform Pliodetes or the lepisosteids.

Another puzzling feature is the presence of a branchiopercle in the opercular series of Ticinolepis. According to Hubbs (1919), in Amia calva the branchiopercle develops in a fold continuous with the lower edge of the lower jaw. In the adult, the branchiopercle of is not only attached to the hyoid arch (anterior ceratohyal), but also to the lower jaw (Grande & Bemis, 1998). The presence of a branchiopercle has so far only been reported in halecomorphs (Grande & Bemis, 1998).

The suspension of the jaw in Ticinolepis is of the ginglymodian type, with single articulation of the lower jaw, which is aided by a distinct quadratojugal. The articulation is double in halecomorphs, in which the quadrate and the symplectic articulate independently with the lower jaw (Grande & Bemis, 1998). In teleosts the articulation is also single via quadrate, but there is no distinct quadratojugal (Patterson, 1973). A distinct splint-like quadratojugal is so far known only in ginglymodians and in some semionotiforms the anterior end of the quadratojugal is fused to the quadrate (López-Arbarello, 2012), as is the case in Ticinolepis gen. nov.

The maxilla in Ticinolepis forms a deep supramaxillary notch that accommodates the supramaxilla. A supramaxillary notch is otherwise only known in the most derived halecomorphs Amia and Cyclurus (Grande & Bemis, 1998). However, the posterior border of the maxilla in Ticinolepis is straight and lacks the notch or concavity which is considered synapomorphic of Halecomorphi. On the other hand, the dentition in the two species of Ticinolepis is of the ginglymodian type, including strong styliform and molariform teeth.

The cleithrum of ginglymodian fishes is characterized by a well-developed medial wing (Grande, 2010),which is also present in Ticinolepis indicating affinities with this clade. The pelvic bone however, most closely resembles the basipterygia of some amiiforms (e.g., Caturus, Amblysemius; Grande & Bemis, 1998), with slender and compressed shaft and triangularly expanded distal end for the articulation with the radials. In the gars, the basipterygial shaft is triangular with proximal end notably broader than the distal end (Grande, 2010). The pelvic bone has been rarely observed in other ginglymodians, but they have been described and illustrated for Macrosemius rostratus (Bartram, 1977: 158, Fig. 16). Although described as similar to those of Amia, the pelvic bones of M. rostratus more closely resemble those of the gars.

As usual, the rhomboid scales in Ticinolepis are gradually smaller starting from the most anterior scales of the lateral line in dorsal, ventral and posterior directions, but in the new genus the ventrum is covered with extremely shallow scales. This feature, which is especially noticeable in the type species, is otherwise only known in some basal halecomorphs and in some crown group palaeoniciformes like Boreosomus and Ptycholepis (A López-Arbarello & T Bürgin, pers. obs., 2015).

Systematics

From the discussion of several anatomical characters in the previous section, it is clear that Ticinolepis shares many features with the Halecomorphi and others with the Ginglymodi. However, the distribution of several of the anatomical traits shared with the halecomorphs (i.e., the features described for the braincase, the frontals with a large postorbital but no antorbital portion, the presence of a descending lamina of the dermopterotic, a branchiopercle or a supramaxillary notch, the distinct squamation of the ventrum) is poorly explored and they have not been included in phylogenetic analyses. The condition in which the ascending process of the parasphenoid extends and meets the sphenotic dorsally is discussed and proposed as a unique feature of halecomorphs by Gardiner and coauthors, but the character is not included in their cladistic analysis of basal neopterygians (Gardiner, Schaeffer & Masserie, 2005: 122 and Appendix S1). Furthermore, according to the most recent hypothesis of halecomorph monophyly, Ticinolepis lacks the synapomorphies of this clade: symplectic involved in a double lower jaw articulation; presence of a notch or concavity in the posterior margin of the maxilla (Grande & Bemis, 1998).

Conversely, the few features that Ticinolepis shares with Ginglymodi have been included in several cladistic analysis and are apomorphic of this clade (the presence of anterior infraorbitals, a medial wing on the cleithrum) or derived only within this clade (the mosaic of suborbitals, the partial fusion of the quadratojugal with the quadrate) (Olsen & McCune, 1991; Grande, 2010; López-Arbarello, 2012; Cavin, Deesri & Suteethorn, 2013). Therefore, based on the current evidence Ticinolepis might be a member of the Ginglymodi. However, the phylogenetic signal of the previously discussed potentially halecomorph features has not been explored yet and, thus, pending a more comprehensive cladistic analysis of crown neopterygian phylogenetic relationships (work in progress by López-Arbarello & Sferco), which is beyond the scope of the present paper, the possible referral of Ticinolepis to the Ginglymodi should be taken as tentative.

Luoxiongichthys hypdorsalis Wen et al., 2012, from the Anisian of China (Guanling formation; Luoping county, Yunnan Province), is also regarded as a neopterygian with a mixture of halecomorphs and ginglymodian characters. Notable differences in body shape and scale ornamentation among other features, clearly distinguish Ticinolepis from Luoxiongichthys (compare with the diagnosis and description of this taxon in Wen et al., 2012). Unfortunately, the most significant potentially halecomorphs features of L. hyperdorsalis, i.e., V-shaped rostral and double jaw joint, are not well documented in the only published study of this fish. One specimen at the Bayerische Staatssammlung für Paläontologie und Geologie (BSPG 2010-I-131) in Munich, Germany, which except for those halecomorphs features closely matches the description of L. hyperdorsalis indicates that this taxon is actually a member of the Ginglymodi (the specimen is included in the phylogenetic analysis of López-Arbarello, 2012). Luoxiongichthys is certainly a very interesting fish and a revision of this taxon is utterly needed to clarify its systematic position.

Intraspecific variation in Ticinolepis longaeva

Compared with the more often discussed examples of sexual dimorphism or traits such as growth and mortality rates or other aspects of the life cycle, intraspecific morphological variation is rarely considered in fish literature. In palaeontology, morphology is the basis of taxonomy and, thus, the analysis of intraspecific morphological variation is essential. However, the small number of sufficiently complete specimens usually hinders such analysis. In the present case, the abundant and well preserved material and the precise stratigraphic and geographic information recorded for each of the studied specimens allows the analysis of different patterns of intraspecific variation in the more widely distributed type species Ticinolepis longaeva.

Morphological changes related to ontogeny are not unusual and some examples have been observed in T. longaeva. The relative size and shape of the preopercle gradually varies among specimens from very slender preopercles with narrowing anteroventral ends in smaller specimens (e.g., MCSN 8413, SL = 115 mm; Fig. 23A) to relatively larger preopercles with lobular anteroventral ends in the larger specimens (e.g., MCSN 8448, SL = 170 mm; Fig. 23B). Additionally, it was mentioned in the description that some of the largest specimens present a pair of small bones between the frontal and the nasals (MCSN 5827, 8469; Figs. 9–10). The complete absence of these bones in the smaller specimens indicate that they only develop in the adults.

Figure 23 Ticinolepis longeva sp. nov., growth related changes in the morphology of the preopercle (painted in green).

(A) Photograph of the skull in MCSN 8413, SL = 115 mm. (B) Photograph of the skull in MCSN 8448, SL = 170 mm. Scale bars = 1 cm.

In the pectoral girdle, the base of the first pectoral ray is poorly developed, does not reach the tip of the base of the other fin rays and narrows towards the base of the second ray in MCSN 8073 (SL = 131 mm; Appendix S1: Fig. 7); it narrows proximally and has a laterally compressed end, but reaches the tip of the base of the second ray in MCSN 8072 (SL = 122 mm; Fig. 7A); it is well developed and broader than the bases of the other fin rays in MCSN 8001 (SL = 197 mm; Fig. 14A). Therefore, although the condition of the first pectoral ray is only visible in a few specimens, it most probably represent another case of ontogenetic variation.

Variation in the total number and relative size of the suborbital bones was also observed and initially thought to be likewise related to ontogeny, apparently due to the fusion of some of the very small and numerous suborbital bones observed in small specimens. However, the presence of a mosaic of very small and numerous suborbitals in one of the largest specimens (PIMUZ T 4987; Appendix S1: Fig. 6) led us to consider other possible explanations. Considering the stratigraphic provenance of the specimens, we concluded that the pattern of suborbitals varies from very small and numerous bones in suborbitals in the specimens from the Besano Formation (e.g., more than >34 PIMUZ T 3269, 44 PIMUZ T 277) to comparatively larger and less numerous suborbitals in the specimens from the Sceltrich beds (e.g., 15 to 20 in specimens MCSN 8467, 8469, 8470) and upper Meride Limestone (e.g., 10 and 15 in MCSN 5827, right and left side respectively) (Fig. 24). The specimens from Cassina show an intermediate state of 25 to 30 suborbitals (counted in the specimens MCSN 8072, 8073, and 8413).

Figure 24 Time related variation in the suborbital bones.

Ticinolepis longeva sp. nov., variation in the amount and relative size of suborbital bones across the stratigraphic column.

Resembling the case described above, the variation in the relative size and number of scales in T. longaeva (analysed through the number of scales above the lateral line along the vertical row immediately anterior the dorsal fin) also showed positive relation with the stratigraphic distribution of the specimens. This relation is evident when comparing specimens of the same size (e.g., SL c. 130 mm in Fig. 25). In this case, however, growth also plays a role determining this variability, at least among the specimens from Cassina (Fig. 25). The latter specimens are indeed very important, not only because they evidence this growth-related trend in the variation of the scales, but also because they fill a range of variation between the older material from the Besano Fm. and the Cava Inferiore, and the younger specimens from the Sceltrich Beds. Without the specimens from Cassina, there would be a devious morphological gap, which together with the gap in the stratigraphic range, would be misleading towards the partition in two different species.

Figure 25 Time related variation in the squamation.

Ticinolepis longeva sp. nov., variation in the number of scales above the lateral line, counted along the vertical row of scales immediately before the origin of the dorsal fin. The number of scales increases with the size of the specimens, but also with the time. Specimens of approximately the same standard length (e.g., c. 130 mm) present less number of scales above the lateral line in the older Besano Formation than in the younger Cassina o Sceltrich Beds of the Meride Limestone.

Empirical data demonstrating time-related morphological variation is only very rarely available in the fossil record. The stratigraphical or paleontological information are usually not detailed enough to allow such analysis, with very few exceptions like the case of the more than 50 species of Semionotus in the Newark Supergroup (McCune, 2004).

Stratigraphic distribution and palaeoecology

The new neopterygian genus Ticinolepis is so far only known from the Middle Triassic sequence of the Monte San Giorgio and includes two species, T. longaeva (type) and T. crassidens spp. nov., which co-occur in the uppermost part of the Besano Formation (lowermost Ladinian), but only the former species has also been found in the overlying lower Ladinian Meride Limestone (Figs. 2–4). Ticinolepis crassidens is thus restricted to the c. 1.30 m outcropping sediments of beds 153 to 176 of the Besano Formation, which represent approximately 80–100 Ka (Furrer, 1995; Stockar, Baumgartner & Condon, 2012). Ticinolepis longaeva has been found from bed number 157 to 180 in the Besano Formation and from the Cava inferiore beds to the uppermost layers of the Meride Limestone (but not in the Kalkschieferzone). This sequence encompasses c. 400 m, representing approximately 2 Myr. Fragmentary material of Ticinolepis sp., which could not be identified to the species level, has been found in the upper part of the middle Besano Formation, which is latest Anisian in age (N. secedensis Ammonoid Zone; Fig. 3). The two species appear more or less simultaneously in the uppermost part of the Besano Formation and co-occur in beds 164, 165, and 175 (Fig. 3). Therefore, there is no evidence for a succession from one species to the other. However, there are indications that T. crassidens is absent in the basin of the Meride Limestone due to paleoenvironmental reasons.

The markedly tritoral dentition, including molariform teeth, fusion of the bones in the lower jaw and the skull roof of Ticinolepis crassidens strongly indicates that this species had an exclusively durophagous diet (Wainwright & Richard, 1995; Bellwood & Hoey, 2004). In fact, T. crassidens represents the oldest example of this trophic specialization in a neopterygian taxon. Prior to this study, the origin of durophagy in neopterygians has been dated in the Late Triassic (Tintori, 1999). In the Norian there are several holostean taxa with an obviously durophagous lifestyle (Lombardo & Tintori, 2005). Also at this time appeared the first pycnodonts, which include many species with molariform teeth and a much higher ecomorphological plasticity (Poyato-Ariza, 2005). In contrast, the dentition of T. longaeva suggests that this species had a more generalist diet and was thus more flexible to adapt to the environmental changes that occurred between the upper part of the Besano Formation and the Meride Limestone. The two species of Ticinolepis thus represent the oldest recorded example of habitat partitioning between sibling species of actinopterygians in the Triassic, which is probably due to a process of sympatric speciation (Dieckmann & Doebeli, 1999; Barluenga et al., 2006; Rocha & Bowen, 2008). Fossil evidence of sympatric speciation is very rare and the best known example for Mesozoic actinopterygians is that of the Semionotus species flock in the Newark Supergroup (McCune, 2004).

The indirect evidence of sympatric speciation in the upper Besano Formation strongly suggest that the corresponding ecosystem had reached a state of equilibrium (Solé, Montoya & Erwin, 2002). The stability of the ecosystem is further indicated by the presence of top predators completing the trophic network. Four different species of Saurichthys occur together with one or both species of Ticinolepis in the beds 153, 162, 163 and 165 of the Besano Formation (Maxwell et al., 2015: Fig. 1). Additionally, the small eosauropterygian reptile Serpianosaurus is very frequent in the lower and upper part of the upper Besano Formation (beds 142–173) and occurs together with one or both species of Ticinolepis in beds 163, 164, 165, 171 and 173 (Rieppel, 1989: 4; H Furrer, 2015, unpublished data).

The upper Besano Formation was deposited in an intraplatform basin with some connections to the open sea (ammonoids are not rare; Rieber, 1973b; Röhl et al., 2001). The Meride Limestone represents a wider but more restricted basin with less connection to the open sea (ammonoids are very rare; Wirz, 1945; Furrer, 1999b; Stockar, 2010). During the time when the upper Besano Formation was deposited the area was adjacent to the shallow platform of the middle San Salvatore Dolomite, where T. crassidens could find hard-shelled food in algal meadows, carbonate sand bars and probably even in small reef-like mounds or along steeper ramps (Zorn, 1971). During deposition of the Meride Limestone, the relief was probably less marked, with a flat ramp to the carbonate platform and reef-like mounds or buildups are not known.

Taking into consideration the fossil assemblage of the upper Besano Formation, possible food for the durophagous Ticinolepis crassidens also includes juvenile ammonoids and small gastropods (Fig. 18A), which are very abundant in this unit (Röhl et al., 2001: page 7, Fig. 4). In contrast, these mollusks are rare components of the fauna of the Meride Limestone. Although T. crassidens is absent in the latter unit, this does not necessarily mean that it went extinct before its deposition; alternatively, this species might simply have inhabited more favourable areas during this time, such as the shallow water platform represented by the middle/upper San Salvatore Dolomite, in which small gastropods e.g., Rasatomaria gentilii (Pieroni & Nützel, 2014), bivalves (mainly Bakevellia and Modiolus), and other small animals living in algal meadows were available (Zorn, 1971). The success of T. crassidens during deposition of the uppermost Besano Formation was probably favoured by the absence of potential competitors with similar strongly tritoral dentitions, such as the species of Colobodus and Crenilepis (Mutter, 2004), which are common from the upper part of the lower to the lower part of the upper Besano Formation, but rare in its uppermost part (Röhl et al., 2001: page 7).

The absence of Ticinolepis longaeva in the Kalkschieferzone, the uppermost unit of the Meride Limestone, might also be due to environmental reasons. As was noticed in the ‘Systematic palaeontology’, the three poorly preserved specimens referred by Bürgin (1995) to Archaeosemionotus sp. from this unit do not belong to the new genus Ticinolepis; they also do not belong to Archaeosemionotus (López-Arbarello, Stockar & Bürgin, 2014) and could not be identified so far. During deposition of the Kalkschieferzone the environment was even more restricted, with a seasonally controlled sedimentation. During this time, a stable density stratification resulted in anoxic bottom waters and increased salinity during dry seasons (partly evaporitic conditions) and periodic freshwater influence during wet seasons (Furrer, 1995). These conditions probably limited the diversity of fishes, mainly dominated by small plankton feeding taxa as Prohalecites and Peltopleurus (Tintori, 1990; Bürgin, 1995).

Conclusions

The revision and anatomical study of more than hundred fish specimens from the Besano Formation and the Meride Limestone, which have previously been identified under the generic name Archaeosemionotus, led to the recognition of the new holostean genus Ticinolepis. Among them, it was not possible to identify 37 specimens beyond the generic level, but the remaining fishes were classified in two new species: T. longaeva (48 specimens) and T. crassidens (20 specimens).

The anatomy of Ticinolepis shows a mosaic of conflicting characters. The fish presents several morphological features that are otherwise only known in halecomorphs, such as the characteristics of the braincase, the ascending process of the parasphenoid in contact with the sphenotic, the large extension of the frontals posteriorly but not anteriorly to the orbit, the presence of a branchiopercle and a supramaxillary notch, and the distinct squamation of the ventrum. However, Ticinolepis also presents two synapomorphies of Ginglymodi (the presence of anterior infraorbitals, a medial wing on the cleithrum) and two other traits derived only within this clade (the mosaic of suborbitals, the partial fusion of the quadratojugal with the quadrate). Based on the current phylogenetic signal of these ginglymodian features, Ticinolepis is here provisionally classified within the Ginglymodi, pending the results of an ongoing cladistic study of crown neopterygians (A López-Arbarello & E Sferco, 2016, unpublished data). The mosaic morphology of Ticinolepis indicates that the fish might represent a stem taxon of Holostei, but it also challenges the monophyly of the two main holostean lineages, in particular that of Ginglymodi.

The morphological analysis of the numerous specimens representing the more widely distributed species Ticinolepis longaeva revealed interesting patterns of intraspecific variation. Some morphological changes occur during ontogeny: relative size and shape of the preopercle and the first pectoral fin ray, the presence of additional small bones between the nasals and the frontals, but other morphological variation have been observed along the stratigraphic succession. These variation has been observed in the relative size and number of suborbital bones and scales.

During the latest Anisian to earliest Ladinian the two species of Ticinolepis coexisted in the intraplatform basin represented by the uppermost Besano Formation, but only T. longaeva inhabited the more restricted basin represented by the Ladinian Meride Limestone, except for its uppermost part (the Kalkschieferzone). The absence of T. crassidens from the Meride Limestone is here interpreted as the consequence of a significant palaeonvironmental change related to the more limited connection to the open sea and the resulting dramatic reduction of food resources for this species (hard-shelled mollusks), which was very specialized on a durophagous diet. On the other hand, both species are absent in the even more restricted and seasonally controlled palaeoenvironment represented by the Kalkschieferzone, which is the uppermost unit of the Meride Limestone. The absence of T. longaeva, with a probably generalist diet, in the Kalkschieferzone is probably the result of marked fluctuations in the physical conditions of the basin due to seasonality.

The partition of habitat by the two species of the new genus Ticinolepis and the trophic specialization of T. crassidens towards a strictly durophagous diet suggest a previous event of sympatric speciation, which together with the presence of top predators indicate well-established ecosystems. Therefore, following the rapid recovery during the Anisian immediately after the end of the coral and coal gaps (Chen & Benton, 2012), the ecosystems of the upper Besano Formation at around 10 Myr after the Permo-Triassic mass extinction had already entered the third phase proposed by Benton et al. (2013) for the recovery of the Triassic biotas, thus providing further evidence favoring the model of ecosystem stepwise recovery pattern (Chen & Benton, 2012).

Supplemental Information

Appendix S1 Supplementary figures

Click here for additional data file.

For their excellent preparation of important specimens TB and HF thank A Ceola, M Hebeisen, H Lanz (all PIMUZ), and C Obrist (Rickenbach BL), RS thanks S Rampinelli, H Lanz, U Oberli, L Zulliger and F Magnani. Specimen MCSN 5827 was provided by D Cook (PIMUZ), and L Heck (PIMUZ) helped with the preparation of Fig. 3. Very specially E Bernard, but also M Richter, and Z Johanson are thanked for access to the collection and assistance during work at the Natural History Museum in London. Thanks are due to E Sferco, K Schröder, O Rauhut and the two reviewers L Cavin and P Brito for enlightening discussions and very helpful suggestions.

Anatomical abbreviations

a.ch anterior ceratohyal bone

a.io anterior infraorbital bone

ag angular bone

ao antorbital bone

a.pl anterior pit line

ap-dpt auto-dermopalatine ossification

b.fu basal fulcrum

bexo basi-exoccipital

bsph basisphenoid

brop branchiopercle

br branchiostegal rays

cl cleithrum

cl.m.w cleithral median wing

cor coronoid bone

cor.t coronoid teeth

crsp cranio-spinal process

d dentary

d.c.fu unpaired basal caudal fulcra

d.cl series of denticles on the cleithrum

dpl dermopalatine bone

dpl.t dermopalatine teeth

dpt dermopterotic bone

dsp dermosphenotic bone

d.t dentary teeth

ecp ectopterygoid bone

enpt endopterygoid bone

epo epiotic

ex extrascapular bone

exo exoccipital

f.hm hyomandibular facet

fhm VII passage of the hyomandibular trunk and the jugular vein

fica foramen for the internal carotid artery

fpsa foramen for the efferent pseudobranchial artery or the posterior palatine branch of the cranial nerve VII

fr frontal bone

fr.fu fringing fulcrum

hhy hypohyal

hm.f foramen and groove for the hyomandibular trunk

hm.op opercular process of the hyomandibula

hm.v.f posteroventrally directed facet of the hyomandibula, probably for the articulation with a medial subopercular process

hym hyomandibula

io infraorbital bone

io.c infraorbital sensory canal

iop interopercle

l.et lateral ethmoid

lj lower jaw

m.c mandibular sensory canal

m.pl middle pit line

mpt metapterygoid bone

mx maxilla

mx.p articular anterior process of the maxilla

na nasal bone

osp orbitosphenoid

op opercle

pa parietal bone

par prearticular

par.t prearticular teeth

p.b.fu paired basal fulcrum

p.ch posterior ceratohyal

pcl postcleithrum

p.d.c.fu paired dorsal caudal fulcrum

pmx premaxilla

pmx.“ap” premaxillary “ascending process ”

pmx.np premaxillary nasal process

pmx.of premaxillary fenestra for the olfactory nerve

pmx.prf premaxillary foramen for the palatine ramus of the facial nerve

pop preopercle

pop.c preopercular sensory canal

pr1 first principal ray

pscl presupracleithrum

psp parasphenoid

psp.ap parasphenoid ascending process

pt.pr posttemporal ventral process

pt posttemporal bone

qu quadrate bone

qj quadratojugal bone

ra radial

rar retroarticular bone

ro rostral bone

sag surangular bone

sc scute

scl supracleithrum

sc.r scale-like fin ray

smx supramaxilla

so.c supraorbital sensory canal

so supraorbital bone

sop subopercle

sp sphenotic

sph sphenotic

spic foramen of the spiracular canal

spt subposttemporal bone

stf subtemporal fossa

su suborbital bone

sy symplectic

t.c temporal sensory canal

u.b.fu unpaired basal fulcrum

vo vomer

X exit of the vagus nerve.

The labels “(r)” or “(l)” after any of the abbreviations indicate right or left elements respectively.

Institutional abbreviations

AMNH American Museum of Natural History, New York, US

MCSN Museo Cantonale di Storia Naturale, Lugano, Switzerland

NHMUK Natural History Museum, London

PIMUZ Paläontologisches Institut und Museum der Universität Zürich, Switzerland.

Additional Information and Declarations

Competing Interests

Author Contributions

Data Availability

New Species Registration

The authors declare there are no competing interests.

Adriana López-Arbarello conceived and designed the experiments, performed the experiments, analyzed the data, wrote the paper, prepared figures and/or tables, reviewed drafts of the paper.

Toni Bürgin performed the experiments, analyzed the data, wrote the paper, prepared figures and/or tables, reviewed drafts of the paper.

Heinz Furrer analyzed the data, contributed reagents/materials/analysis tools, wrote the paper, prepared figures and/or tables, reviewed drafts of the paper.

Rudolf Stockar conceived and designed the experiments, analyzed the data, contributed reagents/materials/analysis tools, wrote the paper, prepared figures and/or tables, reviewed drafts of the paper.

The following information was supplied regarding data availability:

MCSN, Museo Cantonale di Storia Naturale, Lugano, Switzerland: MCSN 4974, 8072, 8001, 8007–9, 8011, 8073, 8077, 8083–4, 8086, 8317, 8351, 8412–13, 8415, 8417, 8424, 8427, 8441, 8446, 8448, 8467, 8468, 8469, 8470, 8471, 8475, 8476, 5827.

PIMUZ, Paläontologisches Institut und Museum der Universität Zürich, Switzerland: PIMUZ-T 226, 237, 240, 248, 252, 256, 257, 265, 273, 275, 276, 277, 278, 315, 331, 341, 358, 362, 401, 413, 438, 486, 638, 1789, 2823, 2835, 2848, 2944, 2999, 3015, 3269, 4350, 4783, 4785, 4788, 4789, 4790, 4987.

NHMUK, Natural History Museum, London: NHMUK P 19351, P 19355.

The following information was supplied regarding the registration of a newly described species:

Ticinolepis gen. nov. urn:lsid:zoobank.org:act:0DDFF9EC-8861-42A5-87A7-744153193A42

Ticinolepis longaeva sp. nov. urn:lsid:zoobank.org:act:31304A6C-ADF6-4C37-92F2-62E505B2823C

Ticinolepis crassidens sp. nov. urn:lsid:zoobank.org:act:87791836-F508-4D3A-97DD-160A32751F01

Publication: urn:lsid:zoobank.org:pub:64547327-C8AE-4CD3-8776-F3CA89E1744E.

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
