# Peer review of "New holostean fishes (Actinopterygii: Neopterygii) from the Middle Triassic of the Monte San Giorgio (Canton Ticino, Switzerland)"

_PeerJ, doi:10.7717/peerj.2234_

## Round 0.1 · original submission · Minor Revisions

I agree with the two reviewers that the manuscript is really well written and complete. It would be nice to see more manuscript of such quality more often!

The reviewers have made some minor suggestions, which I suggest you follow (especially comments of reviewer 2). In addition to the reviewer’s comments, I have noted some minor corrections (see below) that I would like you to correct in the revised version in your manuscript.

I notably noted that many references cited in the text are not referenced at the end of the manuscript. Here is the list: Tintori et al., 2010; López-Arbarello et al., 2011; Wen et al., 2012; Xu & Wu, 2012; Chen et al., 2014; Xu & Chen, 2015; Herzog, 2003; Lombardo & Tintori, 2012; Tintori & Felber, 2015; Agassiz, 1834; Cavin & Suteethorn, 2006; Deesri et al., 2014; Maisey, 2001; Tintori, 1996; Lombardo, Tintori & Tona, 2012; López-Arbarello, 2008; Dieckmann & Doebeli, 1999; Barluenga et al., 2006; Rocha & Bowen, 2008; Pieroni & Nützel, 2014; Tintori, 1990.

Below are additional comments on your manuscript:

- Abstract (last sentence): ‘palaeoenvironmental’, not ‘palaeonevironmental’
- l. 48: ‘early Mesozoic’, not ‘Early Mesozoic’
- l. 52: ‘monophyletic group’ or ‘clade’, but not ‘monophyletic clade’ (a clade is monophyletic by definition)
- l. 279: Müller (1844) is listed as 1846 in the reference list. Which one is correct?
- l. 830: ‘tritorial’ or ‘tritoral’? (check throughout)
- l. 1018: ‘(figure 98)’?
- l. 1022: NHMUK, not BMNHUK
- l. 1031: AMNH should be added to Inst. Abbr.
- l. 1036: NHMUK, not BMNHUK
- l. 1108: Gardiner, Schaeffer & Massery, 1996 is listed as 2005 in the reference list. Which one is correct?
- l. 1179 and 1181: spell out genus name at beginning of sentence
- l. 1371: Bürgin et al. 1991 not cited in the text
- l. 1379: Cope, 1872 not cited in the text
- l. 1536: Scheuring, 1978 not cited in the text
- l. 1542: 2015, not ‘in press’
- Figure 11: could you explain what is figured in B and C in the caption, please?

·

Basic reporting

No Comments.

Experimental design

No Comments.

Validity of the findings

No Comments

Additional comments

I consider that the article "New holostean fishes (Actinopterygii: Neopterygii) from the Middle Triassic of The Monte Giorgio (Canton Ticino, Switzerland)" is an important contribution to the knowledge of Neopterygians and especially for the Holosteans as the taxa examined and described here present a mixture of Gyglymodi and Halecomorph features.
Unfortunately, a formal phylogenetic position of this genus was not yet hypothesized. But the authors make clear that this study is in progress.
So I should acknowledge the excellent work (description, stratigraphy, discussion, figures) and recommend that this article should be accepted for publication as it is.

·

Basic reporting

The paper described a new genus and two new species of a neopterygian on the basis of material previously referred to other taxa. Beyond the description of the material, the study is interesting because the new taxa have characters that question the nature of the holosteans, and the definitions of the ginglymodians and halecomorphs.

Experimental design

The descriptions are very good, and I only did some comments and pointed out some corrections directly in the pdf. In particular, some characters described in the text are sometimes difficult to see on the figures.
The authors did not compare their species with Luoxiongichthys (Wen et al., 2012), a fish from the Anisian of Yunnan, which also has a putative mix of characters from ginglymodians and halecomorphs. I suggest including a short comparison with this taxon.
The authors also detailed the evolution of some characters (squamation, cheek) in the stratigraphical column, which is interesting. They could compare, or at least quote, the study from McCune, which addressed this kind of question within the Semionotus elegans group. It is not so common that the stratigraphical frame allows such kind of analysis.
Caption figure 25: "Time related variation in the squamation". The relation is not with time, but between length and number of scales. Secondly, the analysis of the graph shows that the data are arranged more or less according to time. This is not similar.

Validity of the findings

No comments

Additional comments

I suggest that this article should be published once that the points mentioned above will be fully addressed.

---

## Round 0.2 · accepted · Accept

Thank you for this revised version, which incorporates the different suggestions made by the reviewers and myself. I am pleased to formally accept your manuscript for publication in PeerJ. Congratulations!